# Revisiting the Effect of Topologies on Decentralized SGD

## Abstract

Decentralized SGD is a fundamental algorithm in decentralized learning, although
the influence of an underlying network topology on its convergence behavior is not
yet fully understood. Existing convergence analyses have shown that topologies
with a small spectral gap significantly deteriorate the convergence rate of Decen-
tralized SGD in both homogeneous and heterogeneous cases. However, many
prior papers have reported that indeed the choice of the topology has a significant
experimental impact in the heterogeneous case, but has little experimental impact
on training behavior in the homogeneous case. In this paper, we present a tighter
convergence analysis of Decentralized SGD for both convex and non-convex cases,
offering a more precise understanding of how topologies affect the convergence
rate than the prior analysis. Specifically, unlike existing convergence analyses
that used only the spectral gap as a property of the topology, our novel analysis
shows that all eigenvalues of the mixing matrix affect the convergence rate. This
leads to the key insight that in homogeneous settings, the effect of topology on
the convergence rate is notably smaller than expected from the existing analyses,
especially for commonly used topologies, such as the ring and torus. Throughout
the experiments, we carefully evaluated the convergence behavior of Decentralized
SGD and demonstrated that our novel convergence analysis can more accurately
describe the effect of topology on the convergence rate.

## 1 Introduction

Training machine learning models in a distributed manner has been emerging due to the large-scale
training and privacy preservation (McMahan et al., 2017; Lian et al., 2017; Kairouz et al., 2021).
There are mainly two approaches for distributed learning: federated learning and decentralized
learning. Federated learning assumes that there is a central server and that all nodes are connected
to the central server. However, as the number of nodes increases, federated learning suffers from a
large communication cost between the central server and a large number of nodes, which becomes a
bottleneck in running time. To reduce the communication costs, decentralized learning has gained
significant attention. In decentralized learning, nodes are connected over an underlying network
topology, and each node communicates with its neighbors. Thus, decentralized learning, such as
Decentralized SGD (Lian et al., 2017), is more communication efficient since each node only needs
to exchange parameters with a few number of nodes.

Although Decentralized SGD can reduce the communication costs for each round, its sparse com-
munication characteristic may deteriorate the convergence rate. Specifically, in Decentralized SGD,
each node performs gossip averaging to approximate the average of all nodes' parameters. It is
important to understand how this approximation affects the convergence behavior, and there is a
long line of research studying the convergence rate of Decentralized SGD (Lian et al., 2017; Yu
et al., 2019; Koloskova et al., 2020b; Wang & Joshi, 2021). Then, the existing convergence analysis
showed that sparse topologies significantly degrade the convergence rates in both homogeneous and
heterogeneous cases.

However, many prior papers have reported experimental results that differ from the predictions of
existing convergence analyses of Decentralized SGD (Lian et al., 2017; Luo et al., 2019; Bellet
et al., 2022; Takezawa et al., 2023; Takezawa & Stich, 2025). See Fig. 1 in Bellet et al. (2022) and
Fig. 7 in Takezawa et al. (2023) for the comprehensive experiments on the effect of topology. The

existing convergence analyses typically rely solely on the *spectral gap*—defined via the second-largest eigenvalue in absolute value of the mixing matrix—as the key graph-theoretic quantity controlling convergence (Lian et al., 2017; Koloskova et al., 2020b). They suggest that sparse topologies (with small spectral gap) should result in significantly slower convergence, in both homogeneous and heterogeneous settings. While this aligns with experimental observations in the heterogeneous setting where each node holds a distinct dataset, empirical results consistently show that in the homogeneous setting where all nodes have similar data, the convergence behavior of Decentralized SGD is surprisingly robust to the choice of topology. This discrepancy between theoretical predictions and experimental observation in the homogeneous case has motivated several follow-up studies (Neglia et al., 2020; Vogels et al., 2022; Zhu et al., 2023), but none of them have provided a convergence analysis that accurately accounts for the observed insensitivity to topology. In particular, existing analyses fail to capture the role of the full eigenvalue spectrum of the mixing matrix beyond the spectral gap.

In this paper, we develop a novel convergence analysis of Decentralized SGD that overcomes this limitation by incorporating the influence of *all* eigenvalues of the mixing matrix into the convergence rate, rather than relying solely on the spectral gap. Our new analysis applies to both convex and non-convex objective functions and leads to significantly improved convergence rates, especially in the near-homogeneous regime. We show that the conventional spectral-gap-based convergence rates are overly pessimistic for many commonly-used topologies, such as the ring and torus, and we provide theoretical and numerical evidence that our refined analysis better aligns with observed training behavior. Importantly, our work also has practical implications: prior studies on communication-efficient topology design have focused on improving the spectral gap to accelerate convergence (Chow et al., 2016; Wang et al., 2019; Ying et al., 2021). Our results suggest that the full eigenvalue spectrum plays a more nuanced role in determining convergence and may offer a more effective target for guiding topology design than the spectral gap alone. This insight opens new future directions for developing topologies that can reconcile the convergence rate and communication efficiency.

Our contributions are summarized as follows:

- We develop a novel proof technique and provide a better convergence rate for Decentralized SGD than those shown in previous studies in both convex and non-convex settings.
- Our analysis reveals that the full eigenvalue spectrum of the mixing matrix—not just the spectral gap—governs the convergence rate. This leads to a more accurate characterization of the impact of topology, particularly in near-homogeneous settings.
- We provide experimental evidence showing that our theoretical predictions closely match observed behavior, explaining why sparse topologies with small spectral gaps often perform well in practice despite prior pessimistic analyses.

**Notation:** We write $\|\boldsymbol{x}\|$ for Euclidean norm of vector $\boldsymbol{x}$, $\|\boldsymbol{X}\|_F$ for Frobenius norm of matrix $\boldsymbol{X}$, and $\|\boldsymbol{X}\|_{\mathrm{op}}$ for operator norm. $\mathbf{1}$ is a vector with all ones, and $\mathbf{0}$ is a vector with all zeros.

## 2 PROBLEM SETUP

We consider the following problem, where the loss functions are distributed among $n$ nodes:

$$\min_{\boldsymbol{x} \in \mathbb{R}^d} \left[ f(\boldsymbol{x}) := \frac{1}{n} \sum_{i=1}^{n} f_i(\boldsymbol{x}) \right], \quad f_i(\boldsymbol{x}) := \mathbb{E}_{\xi_i \sim \mathcal{D}_i} \big[ F_i(\boldsymbol{x}; \xi_i) \big],$$

where $\boldsymbol{x}$ denotes the model parameter, $f_i : \mathbb{R}^d \to \mathbb{R}$ denotes the loss function of node $i$, and $\mathcal{D}_i$ denotes the training dataset of node $i$. Let $G = (V, E)$ be the underlying network topology, where $V$ is a set of nodes and $E$ is a set of edges. Let $\boldsymbol{W} \in [0,1]^{n \times n}$ be a mixing matrix associated with $G$. $W_{ij}$ denotes the weight of the edge $(i, j)$, and $W_{ij} > 0$ if and only if $(i, j) \in E$.

### 2.1 ASSUMPTIONS

In this paper, we assume that the following assumptions hold, which are commonly used for analyzing decentralized learning methods (Lian et al., 2017; Koloskova et al., 2020b; Yuan et al., 2022). We first describe the assumption on the underlying network topology.

**Assumption 1.** *The underlying topology is connected, and its mixing matrix $\boldsymbol{W} \in [0,1]^{n \times n}$ is doubly stochastic (i.e., $\boldsymbol{W} \boldsymbol{1} = \boldsymbol{1}$ and $\boldsymbol{W}^\top \boldsymbol{1} = \boldsymbol{1}$) and symmetric. Then, let $\lambda_i$ denote the $i$-th largest eigenvalues of $\boldsymbol{W}$, and $\boldsymbol{W}$ satisfies that $1 = \lambda_1 > \lambda_2 \geq \cdots \geq \lambda_n > -1$.*

When Assumption 1 holds, the following inequality is satisfied.

**Lemma 1.** *Suppose that Assumption 1 holds. For any $\boldsymbol{X} \in \mathbb{R}^{d \times n}$, it holds*

$$\frac{1}{n} \left\| \boldsymbol{X} \boldsymbol{W} - \bar{\boldsymbol{X}} \right\|_F^2 \leq \underbrace{\max_{i \geq 2}(\lambda_i^2)}_{=:1-p} \frac{1}{n} \left\| \boldsymbol{X} - \bar{\boldsymbol{X}} \right\|_F^2, \tag{1}$$

*where $\bar{\boldsymbol{X}} = \frac{1}{n} \boldsymbol{X} \boldsymbol{1} \boldsymbol{1}^\top$ and $\max_{i \geq 2}(\lambda_i^2) < 1$, i.e., $p \in (0,1]$.*

The proof of Lemma 1 is deferred to Section C.3. The above inequality is tight, and the equality holds when $\boldsymbol{X} = (\boldsymbol{v}, \ldots, \boldsymbol{v})^\top$ where $\boldsymbol{v}$ is the eigenvector that corresponds to the second-largest eigenvalue in the absolute value. $p$ is often called the spectral gap, and many prior papers analyzed the convergence rate of decentralized optimization methods by using $p$ to measure how well the underlying topology is connected (Lian et al., 2017; Koloskova et al., 2020b; 2021; Lin et al., 2021; Yuan et al., 2022; Zhao et al., 2022; Di et al., 2024). Generally, in dense topologies, $p$ is large, while in sparse topologies, $p$ approaches zero.

For the loss function, we assume that the following assumptions are satisfied.

**Assumption 2.** $F_i(\cdot, \xi_i)$ *is $L$-smooth for all $i \in \{1, 2, \ldots, n\}$.*

**Assumption 3.** *It holds that $\mathbb{E}[\nabla F_i(\boldsymbol{x}; \xi_i)] = \nabla f_i(\boldsymbol{x})$ for all $\boldsymbol{x} \in \mathbb{R}^d$, and there exists $\sigma \geq 0$ that satisfies $\mathbb{E} \left\| \nabla F_i(\boldsymbol{x}; \xi_i) - \nabla f_i(\boldsymbol{x}) \right\|^2 \leq \sigma^2$ for all $\boldsymbol{x} \in \mathbb{R}^d$ and $i \in \{1, 2, \ldots, n\}$.*

**Assumption 4.** *There exists $\zeta \geq 0$ that satisfies $\frac{1}{n} \sum_{i=1}^n \left\| \nabla f_i(\boldsymbol{x}) - \nabla f(\boldsymbol{x}) \right\|^2 \leq \zeta^2$ for all $\boldsymbol{x} \in \mathbb{R}^d$,*

The heterogeneity $\zeta$ measures how distinct the local loss functions $\{f_i\}_{i=1}^n$ are and increases when each node has a different dataset.

In this paper, we analyze Decentralized SGD in both non-convex and convex cases. For the convex case, it suffices to use Assumptions 6 and 7 instead of Assumptions 3 and 4, as shown in Koloskova et al. (2020b).

**Assumption 5.** *For all $i \in \{1, 2, \ldots, n\}$, $F_i(\cdot, \xi_i)$ is $\mu$-strongly convex with $\mu \geq 0$.*

**Assumption 6.** *Let $\boldsymbol{x}^\star \in \arg\min_{\boldsymbol{x} \in \mathbb{R}^d} f(\boldsymbol{x})$. It holds that $\mathbb{E}[\nabla F_i(\boldsymbol{x}; \xi_i)] = \nabla f_i(\boldsymbol{x})$ for all $\boldsymbol{x} \in \mathbb{R}^d$, and there exists $\sigma_\star \geq 0$ that satisfies $\mathbb{E} \left\| \nabla F_i(\boldsymbol{x}^\star; \xi_i) \right\|^2 \leq \sigma_\star^2$ for all $i \in \{1, 2, \ldots, n\}$.*

**Assumption 7.** *Let $\boldsymbol{x}^\star \in \arg\min_{\boldsymbol{x} \in \mathbb{R}^d} f(\boldsymbol{x})$. Then, there exists $\zeta_\star \geq 0$ that satisfies $\frac{1}{n} \sum_{i=1}^n \left\| \nabla f_i(\boldsymbol{x}^\star) \right\|^2 \leq \zeta_\star^2$.*

Assumptions 3 and 4 suppose that the heterogeneity and stochastic gradient noise are bounded at all parameters, whereas Assumptions 6 and 7 consider only the bound at an optimal parameter, ensuring that $\zeta_\star \leq \zeta$ and $\sigma^\star \leq \sigma$. Note that unlike $\zeta$, $\zeta_\star = 0$ does not necessarily mean that $f_i = f$ for all $i$.

## 2.2 Decentralized SGD

One of the most basic decentralized optimization methods is Decentralized SGD (Lian et al., 2017). In Decentralized SGD, node $i$ updates its parameter $\boldsymbol{x}_i \in \mathbb{R}^d$ as follows:

$$\boldsymbol{x}_i^{(r+1)} = \sum_{j=1}^n W_{ij} \left( \boldsymbol{x}_j^{(r)} - \eta \nabla F_j(\boldsymbol{x}_j^{(r)}; \xi_j^{(r)}) \right), \tag{2}$$

where $\eta > 0$ is the stepsize, and $\xi_j^{(r)}$ corresponds to the minibatch sampled from $\mathcal{D}_j$ in node $j$. Many prior papers have analyzed the convergence rate of Decentralized SGD (Yu et al., 2019; Koloskova et al., 2020b; Le Bars et al., 2023). Under the assumptions we discussed in Section 2.1, the following results are the best convergence rates among the previous studies.

**Proposition 1** (Koloskova et al. (2020b)). *Consider the algorithm shown in Eq. (2). Suppose that $\{\boldsymbol{x}_i^{(0)}\}_{i=1}^n$ are initialized to the same value.*

***Convex Case:*** *Suppose that Assumptions 1, 2 and 5 to 7 hold. Then, there exists $\eta$ such that*

$$\frac{1}{R+1}\sum_{r=0}^{R}(\mathbb{E}f(\bar{\boldsymbol{x}}^{(r)}) - f(\boldsymbol{x}^{\star})) \leq \mathcal{O}\left(\sqrt{\frac{r_0\sigma_\star^2}{nR}} + \left(\left(\frac{\sigma_\star^2}{p} + \frac{\zeta_\star^2}{p^2}\right)\frac{(1-p)Lr_0^2}{R^2}\right)^{\frac{1}{3}} + \frac{Lr_0}{Rp}\right),$$

*where $\bar{\boldsymbol{x}}^{(r)} := \frac{1}{n}\sum_{i=1}^{n}\boldsymbol{x}_i^{(r)}$ and $r_0 := \|\bar{\boldsymbol{x}}^{(0)} - \boldsymbol{x}^{\star}\|^2$.*

***Non-convex Case:*** *Suppose that Assumptions 1, 2, 3 and 4 hold. Then, there exists $\eta$ such that*

$$\frac{1}{R+1}\sum_{r=0}^{R}\mathbb{E}\left\|\nabla f(\bar{\boldsymbol{x}}^{(r)})\right\|^2 \leq \mathcal{O}\left(\sqrt{\frac{L\sigma^2 F_0}{nR}} + \left(\left(\frac{\sigma^2}{p} + \frac{\zeta^2}{p^2}\right)\frac{(1-p)L^2F_0^2}{R^2}\right)^{\frac{1}{3}} + \frac{LF_0}{Rp}\right),$$

*where $\bar{\boldsymbol{x}}^{(r)} := \frac{1}{n}\sum_{i=1}^{n}\boldsymbol{x}_i^{(r)}$ and $F_0 := f(\bar{\boldsymbol{x}}^{(0)}) - \min_{\boldsymbol{x}\in\mathbb{R}} f(\boldsymbol{x})$.*

In both convex and non-convex cases, the spectral gap $p$ appears in the denominator in the second and third terms. For instance, when the topology is a complete graph, torus, and ring, $p$ are 1, $\Omega(n^{-1}), \Omega(n^{-2})$, respectively (Nedić et al., 2018). Since $p$ approaches zero as the topology becomes sparse, Proposition 1 shows that it requires more iterations as the topology becomes sparse in both homogeneous and heterogeneous cases. However, the empirical results reported by many prior papers do not align with this prediction. Many previous studies have experimentally shown that when nodes have similar training datasets (i.e., $\zeta \approx 0$), the training performance is not significantly affected by the topology, even when using a sparse topology such as a ring (Lian et al., 2017; Bellet et al., 2022; Takezawa et al., 2023; Takezawa & Stich, 2025). In contrast, when each node has different datasets (i.e., $\zeta \gg 0$), the choice of topologies significantly affects the training behavior. These observations suggest that the existing convergence analysis does not fully capture the practical training dynamics, particularly when $\zeta \approx 0$ and $\zeta_\star \approx 0$.

## 3 RELATED WORK

**Decentralized Optimization:** One of the most basic decentralized learning methods is Decentralized SGD (Lian et al., 2017), and there is a long line of research attempting to improve this method. Lian et al. (2018); Even et al. (2024) studied asynchronous decentralized learning methods. Nedić & Olshevsky (2016); Assran et al. (2019) studied decentralized learning methods where the underlying network is a directed graph. Lorenzo & Scutari (2016); Nedić et al. (2017); Tang et al. (2018b); Vogels et al. (2021); Lu & De Sa (2021); Di et al. (2024) proposed decentralized learning methods that are robust to data heterogeneity. Li et al. (2020); Sun et al. (2022); Tian et al. (2022) tried to reduce the communication complexity by utilizing the similarity of loss functions that each node has. To further reduce the communication costs, Tang et al. (2018a); Koloskova et al. (2020a); Lu & De Sa (2020); Zhao et al. (2022); Islamov et al. (2025) developed communication compression methods. Chow et al. (2016); Wang et al. (2019); Ying et al. (2021); Song et al. (2022); Ding et al. (2023); Takezawa et al. (2023); You & Pu (2024) developed topologies that reconcile the communication-efficiency and spectral gap to improve the convergence rate of Decentralized SGD.

**Effect of Topology on Decentralized SGD:** One of the most unique procedures of Decentralized SGD is gossip averaging, and it is important to understand how topologies affect the convergence rate. However, as we discussed in Section 2.2, a significant gap exists between theory and practice, and many prior studies have tried to alleviate this issue (Neglia et al., 2020; Vogels et al., 2022; Zhu et al., 2023). Neglia et al. (2020) is the most relevant to our study. They analyzed the convergence rate of Decentralized SGD and showed the effect of eigenvalues other than the spectral gap in the convex case. However, their convergence rates are suboptimal, especially in terms of the number of nodes. Vogels et al. (2022) introduced the novel notion called the effective number of neighbors instead of the spectral gap, analyzing the convergence rate by using the effective number of neighbors. However, they focused on studying the stepsize condition for Decentralized SGD, and the final convergence rate they showed cannot alleviate the gap between theory and experiments that we discussed in Section 2.2. Le Bars et al. (2023); Dandi et al. (2022) introduced a different heterogeneity assumption that depends on the topology, but they cannot improve the convergence rate in the homogeneous case and do not explain why Decentralized SGD is robust to the choice of topology.

# 4 IMPROVED CONVERGENCE RESULTS OF DECENTRALIZED SGD

We now present our novel convergence results of Decentralized SGD and compare them with the existing convergence results. The proofs are deferred to Sections C and D, and the proof sketch is provided in the next Section 5.

## 4.1 MAIN THEOREMS

**Theorem 1.** *Consider the algorithm shown in Eq. (2). Suppose that $\{x_i^{(0)}\}_{i=1}^n$ are initialized to the same value.*

***Convex Case:*** *Suppose that Assumptions 1, 2 and 5 to 7 hold. Then, there exists $\eta$ such that $\frac{1}{R+1}\sum_{r=0}^R (\mathbb{E}f(\bar{x}^{(r)}) - f(x^\star))$ is bounded from above by*

$$\mathcal{O}\left( \sqrt{\frac{r_0\sigma_\star^2}{nR}} + \left( \left( \frac{\sigma_\star^2}{n}\sum_{i=2}^n \frac{\lambda_i^2}{1-\lambda_i^2} + \frac{(1-p)\zeta_\star^2}{p^2} \right) \frac{Lr_0^2}{R^2} \right)^{\frac{1}{3}} + \frac{Lr_0}{Rp} \right),$$

*where $\bar{x}^{(r)} := \frac{1}{n}\sum_{i=1}^n x_i^{(r)}$ and $r_0 := \|\bar{x}^{(0)} - x^\star\|^2$.*

***Non-convex Case:*** *Suppose that Assumptions 1, 2, 3 and 4 holds. Then, there exists $\eta$ such that $\frac{1}{R+1}\sum_{r=0}^R \mathbb{E}\left\|\nabla f(\bar{x}^{(r)})\right\|^2$ is bounded from above by*

$$\mathcal{O}\left( \sqrt{\frac{L\sigma^2 F_0}{nR}} + \left( \left( \frac{\sigma^2}{n}\sum_{i=2}^n \frac{\lambda_i^2}{1-\lambda_i^2} + \frac{(1-p)\zeta^2}{p^2} \right) \frac{L^2 F_0^2}{R^2} \right)^{\frac{1}{3}} + \frac{LF_0}{Rp} \right),$$

*where $\bar{x}^{(r)} := \frac{1}{n}\sum_{i=1}^n x_i^{(r)}$ and $F_0 := f(\bar{x}^{(0)}) - \min_{x\in\mathbb{R}^d} f(x)$.*

In the homogeneous case, we can further improve the convergence rate as follows:

**Theorem 2.** *Consider the algorithm shown in Eq. (2). Suppose that $\{x_i^{(0)}\}_{i=1}^n$ are initialized to the same value.*

***Convex Case:*** *Suppose that Assumptions 1, 2, 5 and 6 hold and $f_i = f_j$ for all $i$ and $j$. Then, there exists $\eta$ such that $\frac{1}{R+1}\sum_{r=0}^R (\mathbb{E}f(\bar{x}^{(r)}) - f(x^\star))$ is bounded from above by*

$$\mathcal{O}\left( \min\left\{ \sqrt{\frac{r_0\sigma_\star^2}{nR}} + \left( \left( \frac{1}{n}\sum_{i=2}^n \frac{\lambda_i^2}{1-\lambda_i^2} \right) \frac{Lr_0^2\sigma_\star^2}{R^2} \right)^{\frac{1}{3}} + \frac{Lr_0}{Rp}, \quad \sqrt{\frac{r_0\sigma_\star^2}{R}} + \frac{Lr_0}{R} \right\} \right),$$

*where $\bar{x}^{(r)} := \frac{1}{n}\sum_{i=1}^n x_i^{(r)}$ and $r_0 := \|\bar{x}^{(0)} - x^\star\|^2$.*

## 4.2 DISCUSSION

$\frac{1}{n}\sum_{i=1}^n \frac{\lambda_i^2}{1-\lambda_i^2}$ **Term:** By comparing Theorem 1 with Proposition 1, the second terms are different. In Proposition 1, only the spectral gap $p(:= 1 - \max_{i\geq 2}(\lambda_i^2))$ is used to measure the property of the topology. In contrast, Theorem 1 shows that eigenvalues $\lambda_2, \lambda_3, \ldots, \lambda_n$ play an important role in the convergence rate and can more precisely describe how the topology affects the convergence rate of Decentralized SGD. From the definition of $p$ in Lemma 1, we have

$$\frac{1-p}{p} = \max_{i\geq 2}\left( \frac{\lambda_i^2}{1-\lambda_i^2} \right) \geq \frac{1}{n}\sum_{i=2}^n \frac{\lambda_i^2}{1-\lambda_i^2}. \tag{3}$$

Thus, the convergence rates shown in Theorems 1 and 2 are better than in Proposition 1. In the above inequality, the equality holds only when $\lambda_2^2 = \lambda_3^2 = \cdots = \lambda_n^2$, and as will be shown in Section 6.1, in commonly-used topologies, such as a ring and torus, $\frac{1}{n}\sum_{i=1}^n(\lambda_i^2/1-\lambda_i^2)$ is considerably smaller than $(1-p)/p$. Thus, in the near-homogeneous case, i.e., $\zeta \approx 0$ or $\zeta_\star \approx 0$, the convergence rate does not deteriorate much even if a sparse topology such as a ring is used as an underlying network. This is also

Table 1: Comparison of transient iterations in the non-convex and homogeneous case. Transient iteration is defined as the number of rounds $R$ required so that $\mathcal{O}(\sqrt{L\sigma^2 F_0/nR})$ becomes dominant in the convergence rate (Ying et al., 2021). The value of $\frac{1}{n}\sum_{i=2}^{n}(\lambda_i^2/1-\lambda_i^2)$ is numerically estimated from Fig. 1. It can be observed that our Theorem 1 improves transient iteration compared to the previous analysis. See Section E for the derivation.

| | Ring | Torus | Hypercube |
|---|---|---|---|
| Proposition 1 (Koloskova et al., 2020b) | $\mathcal{O}(n^7)$ | $\mathcal{O}(n^5)$ | $\mathcal{O}(n^3 \log^2(n))$ |
| Theorem 1 (**ours**) | $\mathcal{O}(n^5)$ | $\mathcal{O}(n^3 \log^2(n))$ | $\mathcal{O}(n^3)$ |

reflected in the comparison shown in Table 1, where Theorem 1 yields a smaller transient iteration compared to Proposition 1. In particular, the improvements are significant for sparse topologies such as the ring. This can provide insight into the experimental results observed in Lian et al. (2017); Bellet et al. (2022); Takezawa et al. (2023), where the impact of topology on convergence rate was less significant than predicted by existing analyses when nodes have similar datasets.

In contrast, the coefficients of the heterogeneity $\zeta$ and $\zeta_\star$ still depend on $p$. As will be explained in Section 5 as a proof sketch, reducing $(1-p)/p$ to $\frac{1}{n}\sum_{i=2}^{n}(\lambda_i^2/1-\lambda_i^2)$ is only possible with respect to $\sigma$. While the coefficient of $\zeta$ and $\zeta_\star$, $(1-p)/p^2$, may be marginally improved under alternative heterogeneity assumptions considered by Le Bars et al. (2023); Dandi et al. (2022), the dependence on the spectral gap appears to be inevitable. This aligns well with the experimental observation shown in previous studies (Bellet et al., 2022; Takezawa et al., 2023), which indicates that the choice of topologies significantly affects the training performance when data distributions vary across nodes.

$\sqrt{\frac{r_0\sigma^2}{R}} + \frac{Lr_0}{R}$ **Term:** We next focus on the homogeneous case and compare Theorem 2 with Proposition 1. When the underlying topology is disconnected, we have $p = 0$ and $\lambda_2 = 1$. Proposition 1 showed that Decentralized SGD does not converge in both convex and non-convex cases, whereas Theorem 2 showed that Decentralized SGD converges to the optimal solution in the convex and homogeneous case. When $f$ is convex, we have $f(\bar{x}) \leq \frac{1}{n}\sum_{i=1}^{n} f(x_i)$. Thus, Decentralized SGD cannot achieve the linear speedup, but it can converge to the optimal solution with the same convergence rate as SGD. A similar discussion can be found in the studies on Local SGD (Woodworth et al., 2020; Glasgow et al., 2022; Patel et al., 2024). Note that this property is specific to convex functions; in the non-convex case, such convergence is not guaranteed, and Decentralized SGD may fail to converge since different nodes can converge to different stationary points, and the average parameter is not necessarily a stationary point.

**Comparison with Other Analyses of Decentralized SGD:** Several prior papers have attempted to capture the influence of topologies using quantities other than the spectral gap (Neglia et al., 2020; Vogels et al., 2022). In the following, we compare our results with theirs. See Section A for more details. The paper most closely related to ours is Neglia et al. (2020), which showed the rate depending on all eigenvalues of $W$. However, they used the strong assumption that the stochastic gradient is bounded instead of assuming that $F_i$ is smooth, and their results failed to improve the convergence rate shown in Proposition 1. Vogels et al. (2022) used the novel notion called the effective number of neighbors instead of the spectral gap and analyzed the convergence rate. However, the effective number of neighbors depends on a hyperparameter, and it is unclear whether their rate is better than the rate shown in Proposition 1. Therefore, Theorem 1 is the first result that improves the well-known convergence rate shown in Proposition 1 without using the additional assumptions, thereby capturing more precisely how topologies influence the convergence rate.

## 5 PROOF SKETCH

In this section, we provide the proof sketch for the non-convex case of Theorem 1. In the following, we use the notations $X \in \mathbb{R}^{d \times n}$, $\nabla F(X; \xi) \in \mathbb{R}^{d \times n}$, and $\nabla F(X) \in \mathbb{R}^{d \times n}$ defined as follows:

$$X^{(r)} := \left(x_1^{(r)}, \ldots, x_n^{(r)}\right), \quad \nabla F(X^{(r)}; \xi^{(r)}) := \left(\nabla F_1(x_1^{(r)}; \xi_1^{(r)}), \ldots, \nabla F_n(x_n^{(r)}; \xi_n^{(r)})\right),$$

$$\nabla F(X^{(r)}) := \left(\nabla f_1(x_1^{(r)}), \ldots, \nabla f_n(x_n^{(r)})\right).$$

**Key Lemma:** The most important lemma for improving the convergence rate is the following.

**Lemma 2.** *Suppose that Assumptions 1 and 3 hold. It holds that*

$$\frac{1}{n}\mathbb{E}\left\|\left(\nabla F(\boldsymbol{X}^{(r)};\xi^{(r)}) - \nabla F(\boldsymbol{X}^{(r)})\right)\left(\boldsymbol{W} - \frac{1}{n}\mathbf{1}\mathbf{1}^\top\right)\right\|_F^2 \leq \frac{\sigma^2}{n}\sum_{i=2}^{n}\lambda_i^2. \tag{4}$$

*Note that $\frac{1}{n}\sum_{i=2}^{n}\lambda_i^2 \leq \max_{i\geq 2}(\lambda_i^2) < 1$.*

As shown in Lemma 1, gossip averaging can reduce the consensus error $\|\boldsymbol{X} - \bar{\boldsymbol{X}}\|_F^2$ by only a factor of $1 - p$ if we consider the worst $\boldsymbol{X}$ in Eq. (1). However, in Eq. (4), $\nabla F_1(\boldsymbol{x}_1, \xi_1) - \nabla f_1(\boldsymbol{x}_1), \nabla F_2(\boldsymbol{x}_2, \xi_2) - \nabla f_2(\boldsymbol{x}_2), \ldots, \nabla F_n(\boldsymbol{x}_n, \xi_n) - \nabla f_n(\boldsymbol{x}_n)$ are independent and have the same mean of zero. Focusing on this property, we can reduce to $\max_{i\geq 2}(\lambda_i^2)$ to $\frac{1}{n}\sum_{i=2}^{n}\lambda_i^2$, as shown in Lemma 2.

**Proof Sketch:** When Assumptions 1 to 4 hold and $\eta \leq \frac{1}{4L}$, Decentralized SGD satisfies the following inequality (see the proof in Lemma 17):

$$\frac{1}{4(R+1)}\sum_{r=0}^{R}\mathbb{E}\left\|\nabla f(\bar{\boldsymbol{x}}^{(r)})\right\|^2 \leq \frac{f(\bar{\boldsymbol{x}}^{(0)}) - f^\star}{\eta(R+1)} + \frac{L\sigma^2}{n}\eta + \frac{L^2}{R+1}\sum_{r=0}^{R}\Xi^{(r)}, \tag{5}$$

where $\Xi^{(r)} := \frac{1}{n}\mathbb{E}\|\boldsymbol{X}^{(r)} - \bar{\boldsymbol{X}}^{(r)}\|^2$ and $\bar{\boldsymbol{X}} := \frac{1}{n}\boldsymbol{X}\mathbf{1}\mathbf{1}^\top$. The above inequality is almost identical to the inequality that often appears in the convergence analysis of SGD, except for the consensus error $\Xi$. Thus, the convergence rate deteriorates if the parameters held by each node are far from the average. By carefully analyzing $\Xi$, we can obtain the following inequality:

$$\Xi^{(r+1)} \leq \left(1 + \frac{p}{2}\right)\frac{1}{n}\mathbb{E}\left\|\boldsymbol{X}^{(r)}\boldsymbol{W} - \bar{\boldsymbol{X}}^{(r)}\right\|_F^2 + \frac{p(1-p)}{4}\Xi^{(r)}$$

$$+ \frac{6\zeta^2}{p}(1-p)\eta^2 + \frac{\eta^2}{n}\mathbb{E}\left\|\left(F(\boldsymbol{X}^{(r)};\xi^{(r)}) - \nabla F(\boldsymbol{X}^{(r)})\right)\left(\boldsymbol{W} - \frac{1}{n}\mathbf{1}\mathbf{1}^\top\right)\right\|_F^2, \tag{6}$$

when $\eta \leq \frac{p}{5L}$ (See the proof of Lemma 3 with $k = 0$). The first and second terms are the remaining consensus error in the previous round. Then, if the third and fourth terms are zero, we can show that $\Xi$ decreases consistently since $\frac{1}{n}\mathbb{E}\|\boldsymbol{X}^{(r)}\boldsymbol{W} - \bar{\boldsymbol{X}}^{(r)}\|_F^2 \leq (1-p)\Xi^{(r)}$. Thus, only the third and fourth terms attempt to make the parameters held by nodes drift away in each round, and as we discussed above, the fourth term can be bounded by Eq. (4).

Using Eq. (6) and Lemma 2, we would like to derive the upper bound of $\Xi^{(r)}$ to obtain the convergence rate. However, Eq. (6) is not a simple recursive inequality since it contains $\mathbb{E}\|\boldsymbol{X}\boldsymbol{W} - \bar{\boldsymbol{X}}\|_F^2$ in the right-hand side, which makes it difficult to obtain the upper bound of $\Xi$. To alleviate this issue, we derive the following lemma.

**Lemma 3.** *Suppose that Assumptions 1 to 4 hold. Then, when $\eta \leq \frac{p}{5L}$, it holds that*

$$\Xi^{(r+1,k)} \leq \left(1 + \frac{p}{2}\right)\Xi^{(r,k+1)} + \frac{p}{4}(1-p)^{k+1}\Xi^{(r,0)} + \frac{6\zeta^2\eta^2}{p}(1-p)^{k+1} + \frac{\sigma^2\eta^2}{n}\sum_{i=2}^{n}\lambda_i^{2(k+1)}, \tag{7}$$

*where $\Xi^{(r,k)} := \frac{1}{n}\mathbb{E}\left\|\boldsymbol{X}^{(r)}\boldsymbol{W}^k - \bar{\boldsymbol{X}}^{(r)}\right\|_F^2$.*

Note that Eq. (7) contains almost the same inequality as Eq. (6) when $k = 0$. In Eq. (7), $\Xi^{(r,k+1)}$ and $\Xi^{(r,0)}$ appear in the right-hand side, while both terms can be bounded from above by using the same inequality, Eq. (7). Thus, recursively applying Eq. (7) yields the following lemma.

**Lemma 4.** *Suppose that Assumptions 1 to 4 hold, and $\{\boldsymbol{x}_i^{(0)}\}_{i=1}^{n}$ are initialized to the same value. When $\eta \leq \frac{p}{5L}$, it holds that*

$$\Xi^{(r)} = \Xi^{(r,0)} \leq \frac{24\zeta^2(1-p)}{p^2}\eta^2 + 3\sigma^2\left(\frac{1}{n}\sum_{i=2}^{n}\frac{\lambda_i^2}{1-\lambda_i^2}\right)\eta^2. \tag{8}$$

Combining Eq. (5) and Lemma 4 and tuning the stepsize $\eta$, we can obtain the convergence rate shown in Theorem 1.

In summary, the convergence rate of Decentralized SGD worsens as the parameters each node has drift away, and the consensus error $\Xi$ increases by the stochastic gradient noise, heterogeneity, and the inexact averaging of the gossip averaging. The existing analysis used the spectral gap $p$ to measure the inexactness of gossip averaging, but the stochastic gradient noises of each node are independent of each node. Indeed, in the worst-case scenario, the spectral gap $p$ is used to bound the error of gossip averaging, as shown in Lemma 1. However, by leveraging the independence of stochastic gradient noise across nodes, we can refine this bound and reduce the dependence from $(1-p)/p$ to $\frac{1}{n}\sum_{i=2}^{n}\left(\lambda_i^2/1-\lambda_i^2\right)$, which captures the contribution of all eigenvalues rather than relying solely on the spectral gap, i.e., the second-largest eigenvalue in the absolute value.

## 6 NUMERICAL EVALUATION

### 6.1 COMPARISON OF $\frac{1-p}{p}$ AND $\frac{1}{n}\sum_{i=1}^{n}\frac{\lambda_i^2}{1-\lambda_i^2}$

In this section, we numerically compared $(1-p)/p$ and $\frac{1}{n}\sum_{i=2}^{n}\left(\lambda_i^2/1-\lambda_i^2\right)$ of commonly used topologies. We depicted the results in Fig. 1. The results indicate that $\frac{1}{n}\sum_{i=2}^{n}\left(\lambda_i^2/1-\lambda_i^2\right)$ is significantly smaller than $(1-p)/p$. For instance, $(1-p)/p$ increases quadratically and linearly, whereas $\frac{1}{n}\sum_{i=2}^{n}\left(\lambda_i^2/1-\lambda_i^2\right)$ increases only linearly and logarithmically, for the ring and torus, respectively. This can explain why the topologies have less experimental impact than what was anticipated by the existing convergence analysis in the homogeneous case.

### 6.2 EFFECT OF $\frac{1}{n}\sum_{i=1}^{n}\frac{\lambda_i^2}{1-\lambda_i^2}$ ON DECENTRALIZED SGD

In this section, we evaluated the convergence behavior of Decentralized SGD by varying topologies and demonstrated that all eigenvalues of the mixing matrix affect the convergence rate.

**Experimental Setup:** We set the number of nodes $n$ to 200, used MNIST (Lecun et al., 1998) as a training dataset, and set the minibatch size to one. To evaluate the behavior of Decentralized SGD in the homogeneous setting, we distributed the training dataset to nodes so that all nodes share

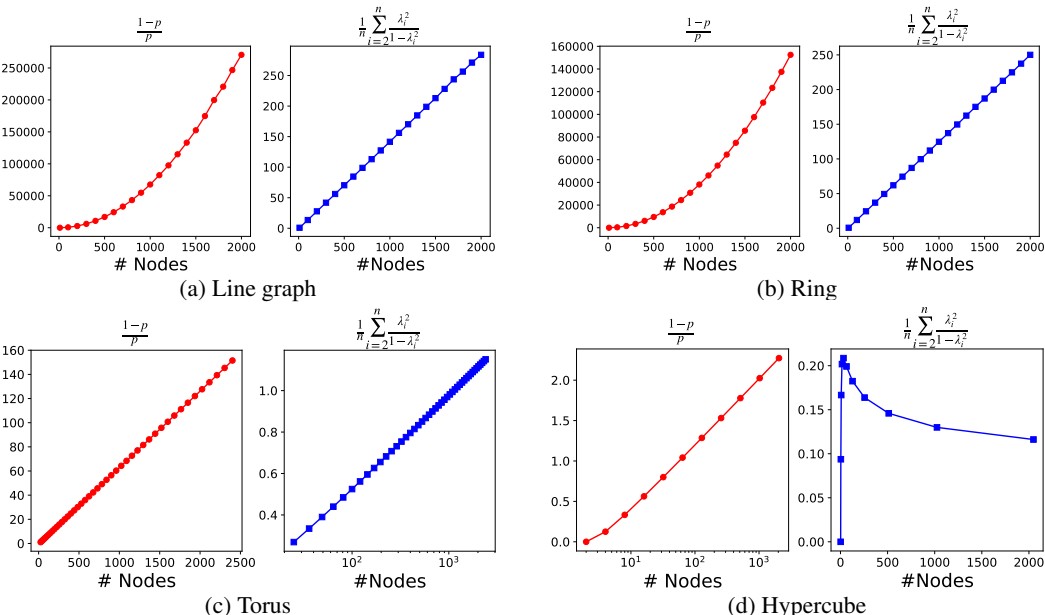

(a) Line graph      (b) Ring

(c) Torus      (d) Hypercube

Figure 1: Comparision between $(1-p)/p$ and $\frac{1}{n}\sum_{i=2}^{n}\left(\lambda_i^2/1-\lambda_i^2\right)$ for commonly used topologies. We used the Metropolis weights (Xiao et al., 2005) for the mixing matrix $\boldsymbol{W}$. Note that we use a logarithmic scale on the x-axis for the torus and hypercube. It can be observed that $\frac{1}{n}\sum_{i=2}^{n}\left(\lambda_i^2/1-\lambda_i^2\right)$ is considerably smaller than $(1-p)/p$.

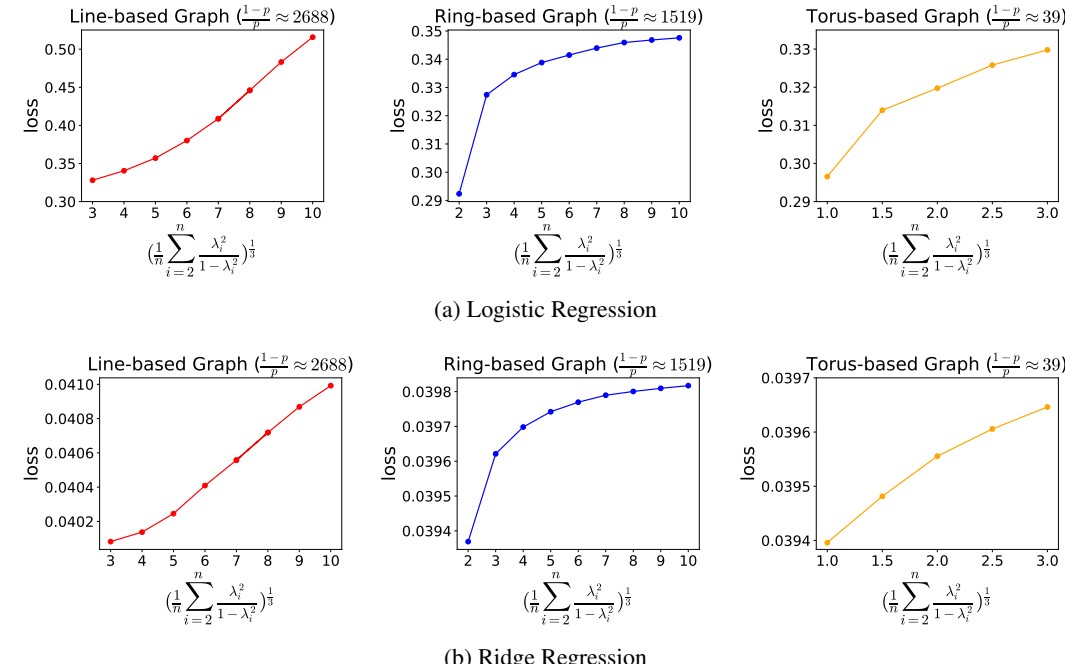

(a) Logistic Regression

(b) Ridge Regression

Figure 2: The impact of $\frac{1}{n}\sum_{i=2}^{n}\big(\lambda_i^2/1-\lambda_i^2\big)$ on the loss value at the final parameter. In each figure, we vary the eigenvalues to construct different topologies while keeping $p$ constant. We observe that the loss value increases as $\frac{1}{n}\sum_{i=2}^{n}\big(\lambda_i^2/1-\lambda_i^2\big)$ increases, which is consistent with Theorems 1 and 2. Error bars are omitted since all standard errors were smaller than $1.0 \times 10^{-5}$ and visually indistinguishable.

the same training dataset. We constructed an underlying graph, such as a ring, line, and torus, and calculated its eigenvalues and eigenvectors. We then altered the eigenvalues, except for the first and second largest eigenvalues in absolute value, and reconstructed various graphs with different $\frac{1}{n}\sum_{i=2}^{n}\big(\lambda_i^2/1-\lambda_i^2\big)$ but the same $p$. See Section F for more details. Using these graphs, we evaluated how the convergence behavior of Decentralized SGD is affected by graphs. We tuned the stepsize by grid search over $\{0.04, 0.05, \ldots, 0.1, 0.2\}$ so that the final loss value was minimized in each experiment. All experiments were repeated with different seed values three times, and we reported the average. We conducted all experiments on the server with Intel Xeon CPU E7-8890 v4.

**Results:** We showed the results in Fig. 2. The results indicate that even if $p$ is the same, the loss values decrease as $\frac{1}{n}\sum_{i=2}^{n}\big(\lambda_i^2/1-\lambda_i^2\big)$ decreases. This observation is consistent with our statement in Theorems 1 and 2 that not only the second largest eigenvalue in absolute value, but also all eigenvalues, affect the convergence rate.

## 7 Conclusion

In this paper, we develop a novel proof technique and provide a better convergence rate for Decentralized SGD than that shown in the prior papers in both convex and non-convex settings. Our novel convergence rates can describe how topologies affect the convergence rate of Decentralized SGD more accurately than the existing rates. Specifically, previous analyses relied solely on the spectral gap to assess the topology effect, suggesting that convergence rates depend only on the spectral gap in both homogeneous and heterogeneous cases. In contrast, our novel analysis shows that all eigenvalues of the mixing matrix play an important role in the convergence rate. Then, we show that for commonly-used topologies, such as a ring and torus, our convergence rates are significantly better than those shown in the existing papers and show that topologies are provably less impactful than predicted by them, especially when nodes have similar datasets.

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

# A    DETAILED COMPARISON WITH PRIOR CONVERGENCE ANALYSIS

In this section, we show the convergence rates derived in Neglia et al. (2020) and Vogels et al. (2022) and compare them with Theorem 1.

**Proposition 2** (Proposition 3.1 in Neglia et al. (2020)). *Suppose that $F_i(\cdot; \xi_i)$ is convex, $\boldsymbol{x}_i^{(0)} = \boldsymbol{x}^{(0)}$ for all $i$, and Assumption 1 holds. Then, using orthogonal projection matrix $P_i$, we rewrite the mixing matrix $\boldsymbol{W}$ as follows:*

$$\boldsymbol{W} = \sum_{i=1}^{n} \lambda_i \boldsymbol{P}_i.$$

*Under these assumptions. there exists a stepsize $\eta$ that satisfies*

$$\mathbb{E}[f(\tilde{\boldsymbol{x}}^{(R)})] - f(\boldsymbol{x}^{\star}) \leq \mathcal{O}\left( \frac{n\|\boldsymbol{x}^{(0)} - \boldsymbol{x}^{\star}\|^2}{\eta R} + \eta E + \eta H \sqrt{E_{sp}} \left( (1 - \alpha)\frac{R - 1}{R} + \frac{\alpha}{1 - \max_{i \geq 2} |\lambda_i|} \right) \right),$$

*where*

$$\tilde{\boldsymbol{x}}^{(R)} := \frac{1}{n(R+1)} \sum_{r=0}^{R} \sum_{i=1}^{n} \boldsymbol{x}_i^{(r)},$$

$$E := \sup_{\{\boldsymbol{x}_i\}_i} \sum_{i=1}^{n} \mathbb{E} \|\nabla F_i(\boldsymbol{x}_i; \xi_i)\|^2,$$

$$H := \sup_{\{\boldsymbol{x}_i\}_i} \sum_{i=1}^{n} \mathbb{E} \|\nabla F_i(\boldsymbol{x}_i; \xi_i)\|,$$

$$E_{sp} := \sup_{\{\boldsymbol{x}_i\}_i} \sum_{i=1}^{n} \mathbb{E} \left\| \nabla F_i(\boldsymbol{x}_i; \xi_i) - \frac{1}{n} \sum_{j=1}^{n} \nabla F_j(\boldsymbol{x}_j; \xi_j) \right\|^2,$$

$$\alpha := \begin{cases} 1 & \text{if } \max_{i \geq 2} |\lambda_i| > 0 \\ \sqrt{\sum_{i=2}^{n} e_i |\frac{\lambda_i}{\max_{i \geq 2} |\lambda_i|}|} & \text{otherwise} \end{cases},$$

*and $e_i$ is an upper bound for the normalized fraction of $\sum_{i=1}^{n} \mathbb{E} \left\| \nabla F_i(\boldsymbol{x}_i; \xi_i) - \frac{1}{n} \sum_{j=1}^{n} \nabla F_j(\boldsymbol{x}_j; \xi_j) \right\|^2$ in the subspace defined by $\boldsymbol{P}_i$*

**Proposition 3** (Theorem 1 in Vogels et al. (2022)). *Suppose that Assumptions 2, 5 and 6 hold, $f_i = f$, and $\boldsymbol{x}_i^{(0)} = \boldsymbol{x}^{(0)}$ for all $i$. Then, there exists a stepsize $\eta$ that satisfies*

$$\left\| \boldsymbol{X}^{(r)} - \boldsymbol{X}^{\star} \right\|_{\boldsymbol{M}}^2 \leq (1 - \mu\eta)^r \left\| \boldsymbol{X}^{(0)} - \boldsymbol{X}^{\star} \right\|_{\boldsymbol{M}}^2 + \frac{8\eta\sigma^2}{n_{\boldsymbol{W}}(\gamma)},$$

*where*

$$\boldsymbol{X}^{(r)} := \left( \boldsymbol{x}_1^{(1)}, \cdots, \boldsymbol{x}_n^{(r)} \right),$$

$$\boldsymbol{M} := (1 - \gamma)\boldsymbol{W}^2 (\boldsymbol{I} - \gamma\boldsymbol{W}^2)^{-1},$$

$$n_{\boldsymbol{W}}(\gamma) := \frac{\frac{1}{1-\gamma}}{\frac{1}{n} \sum_{i=1}^{n} \frac{\lambda_i^2}{1 - \gamma\lambda_i^2}}.$$

The convergence rate shown in Neglia et al. (2020) requires that the stochastic gradient is bounded, i.e., $\|\nabla F_i(\boldsymbol{x}; \xi_i)\| \leq G$ for any $\boldsymbol{x}$, to ensure the term in the right-hand side diminishes as $R$ increases, and Neglia et al. (2020) fails to improve the rate shown in Proposition 1. The analysis shown in Vogels et al. (2022) uses the novel quantity $n_{\boldsymbol{W}}(\gamma)$ instead of the spectral gap. However, as $n_{\boldsymbol{W}}(\gamma)$ depends on the hyperparameter $\gamma$, it is unclear whether this rate is better than the rate shown in Proposition 1. Therefore, Theorem 1 is the first result that improves the well-known convergence rate shown in Proposition 1 without using the additional assumptions.

## B FUTURE WORK

In this section, we discuss the potential application of Theorems 1 and 2. One of the most impressive applications is to develop topologies that minimize $\frac{1}{n}\sum_{i=1}^{n}(\lambda_i^2/1-\lambda_i^2)$. To design topologies that can improve the convergence rate, the topologies with large spectral gap have been extensively studied by Chow et al. (2016); Wang et al. (2019); Ying et al. (2021); Song et al. (2022); Ding et al. (2023); Takezawa et al. (2023); You & Pu (2024). However, our novel analysis discovers that in the near-homogeneous case, the dependence of the spectral gap can be reduced to $\frac{1}{n}\sum_{i=1}^{n}(\lambda_i^2/1-\lambda_i^2)$. Minimizing $(1-p)/p$ also makes $\frac{1}{n}\sum_{i=1}^{n}(\lambda_i^2/1-\lambda_i^2)$ smaller, but if we can design a topology that directly minimizes $\frac{1}{n}\sum_{i=1}^{n}(\lambda_i^2/1-\lambda_i^2)$, we may be able to further improve the convergence rate and make training more stable. For instance, we propose to generalize the approach of Wang et al. (2019) to maximize our new quantity $\frac{1}{n}\sum_{i=2}^{n}\frac{\lambda_i^2}{1-\lambda_i^2}$, instead of maximizing the spectral gap. More precisely, Wang et al. (2019) proposed to decompose the original topology into matching in such a way that the spectral gap is maximized. That allows at the same time to improve the spectral gap of the topology, and use less communications due to alternating between different matchings, providing improvement not only in the convergence speed, but also improving the runtime per iteration. We therefore propose to replace the maximization metric from the spectral gap to our new quantity $\frac{1}{n}\sum_{i=2}^{n}\frac{\lambda_i^2}{1-\lambda_i^2}$, capturing tighter the practical behavior of the decentralized learning algorithms.

Other promising further directions include exploring the extension of our novel theorems to other decentralized optimization algorithms. Although this paper focuses on the analysis of Decentralized SGD, our new proof techniques described in Section 5 are more general and may be applicable in the analysis of other decentralized optimization algorithms, such as Gradient Tracking (Lorenzo & Scutari, 2016; Nedić et al., 2017; Koloskova et al., 2021). Furthermore, it would be interesting to extend Theorems 1 and 2 to the case of gossip averaging with time-varying topologies (Koloskova et al., 2020b) and accelerated gossip averaging (Liu & Morse, 2011; Di et al., 2024).

# C  PROOF OF THEOREM 1

## C.1  INTUITION BEHIND THE IMPROVED ANALYSIS

The spectral gap $1 - \max_{i \geq 2}(\lambda_i^2)$ satisfies the following inequality for any matrix $\mathbf{X}$ (see Lemma 1):

$$\|\mathbf{X}\mathbf{W} - \bar{\mathbf{X}}\|_F^2 \leq \max_{i \geq 2}(\lambda_i^2)\|\mathbf{X} - \bar{\mathbf{X}}\|_F^2$$

On the other hand, if we restrict the input $X := (X_1, X_2, \ldots, X_n)$ so that $X_1, \ldots, X_n$ are **independent** random variables, have the same mean, and their variance is bounded by $\sigma^2$, we have (see Lemma 6):

$$\mathbb{E}\|X\mathbf{W} - \bar{X}\|_F^2 \leq \sigma^2 \sum_{i=2}^n \lambda_i^2$$

*The stochastic noise is independent among nodes.* Thus, the coefficient of the stochastic noise $\sigma^2$ can be reduced to $\frac{1}{n}\sum_{i=2}^n \frac{\lambda_i^2}{1-\lambda_i^2}$ from $\frac{1-p}{p} := \max_{i \geq 2}(\frac{\lambda_i^2}{1-\lambda_i^2})$. This is an intuition of why the coefficient of $\sigma^2$ is improved in our theorem.

## C.2  NOTATION

Define $\boldsymbol{X}, \boldsymbol{X}^\star, \nabla f(\boldsymbol{X}), \nabla F(\boldsymbol{X}; \xi) \in \mathbb{R}^{d \times n}$ as follows:

$$\boldsymbol{X}^{(r)} := \left(\boldsymbol{x}_1^{(r)}, \ldots, \boldsymbol{x}_n^{(r)}\right), \quad \boldsymbol{X}^\star := (\boldsymbol{x}^\star, \ldots, \boldsymbol{x}^\star), \quad \nabla f(\boldsymbol{X}^{(r)}) := \left(\nabla f(\boldsymbol{x}_1^{(r)}), \ldots, \nabla f(\boldsymbol{x}_n^{(r)})\right),$$

$$\nabla F(\boldsymbol{X}^{(r)}) := \left(\nabla f_1(\boldsymbol{x}_1^{(r)}), \ldots, \nabla f_n(\boldsymbol{x}_n^{(r)})\right),$$

$$\nabla F(\boldsymbol{X}^{(r)}; \xi^{(r)}) := \left(\nabla F_1(\boldsymbol{x}_1^{(r)}; \xi_1^{(r)}), \ldots, \nabla F_n(\boldsymbol{x}_n^{(r)}; \xi_n^{(r)})\right).$$

Using the above notations, we can rewrite the update rule shown in Eq. (2) as follows:

$$\boldsymbol{X}^{(r+1)} = \left(\boldsymbol{X}^{(r)} - \eta \nabla F(\boldsymbol{X}^{(r)}; \xi^{(r)})\right) \boldsymbol{W}.$$

In the following sections, we define $\pm a := a - a$ for all $a$. $\mathbb{E}[\cdot]$ denotes the expectation over all randomness during the training, and $\mathbb{E}_r[\cdot]$ denotes the expectation over randomness that occurs at round $r$.

## C.3  USEFUL LEMMAS

**Lemma 5.** *Suppose that Assumption 1 holds. Then, the eigenvalues of $\boldsymbol{W} - \frac{1}{n}\mathbf{1}\mathbf{1}^\top$ are $0, \lambda_2, \lambda_3, \ldots, \lambda_n$.*

*Proof.* Since we have $\boldsymbol{W}\mathbf{1} = \mathbf{1}$, $\mathbf{1}$ is an eigenvector that corresponds to eigenvalue $\lambda_1(= 1)$. Since we have

$$\left(\boldsymbol{W} - \frac{1}{n}\mathbf{1}\mathbf{1}^\top\right)\mathbf{1} = \mathbf{0},$$

$\boldsymbol{W} - \frac{1}{n}\mathbf{1}\mathbf{1}^\top$ has an eigenvalue of $0$.

Let $\boldsymbol{v}_2, \ldots, \boldsymbol{v}_n \in \mathbb{R}^n$ be the eigenvectors of $\boldsymbol{W}$ corresponding to $\lambda_2, \ldots, \lambda_n$, respectively. Since $\lambda_1 \neq \lambda_i$ for all $i \geq 2$, it holds that $\boldsymbol{v}_i^\top \mathbf{1} = 0$. We have

$$\left(\boldsymbol{W} - \frac{1}{n}\mathbf{1}\mathbf{1}^\top\right)\boldsymbol{v}_i = \boldsymbol{W}\boldsymbol{v}_i = \lambda_i \boldsymbol{v}_i.$$

Thus, $\lambda_2, \ldots, \lambda_n$ are eigenvalues of $\boldsymbol{W} - \frac{1}{n}\mathbf{1}\mathbf{1}^\top$. $\qquad\square$

**Lemma 1.** *Suppose that Assumption 1 holds. For any $\boldsymbol{X} \in \mathbb{R}^{d \times n}$, it holds*

$$\frac{1}{n} \left\| \boldsymbol{X}\boldsymbol{W} - \bar{\boldsymbol{X}} \right\|_F^2 \leq \underbrace{\max_{i \geq 2}(\lambda_i^2)}_{=:1-p} \frac{1}{n} \left\| \boldsymbol{X} - \bar{\boldsymbol{X}} \right\|_F^2, \tag{1}$$

*where $\bar{\boldsymbol{X}} = \frac{1}{n}\boldsymbol{X}\mathbf{1}\mathbf{1}^\top$ and $\max_{i \geq 2}(\lambda_i^2) < 1$, i.e., $p \in (0, 1]$.*

*Proof.* We have

$$\begin{aligned}
\left\| \boldsymbol{X}\boldsymbol{W} - \bar{\boldsymbol{X}} \right\|_F^2 &= \left\| (\boldsymbol{X} - \bar{\boldsymbol{X}})(\boldsymbol{W} - \frac{1}{n}\mathbf{1}\mathbf{1}^\top) \right\|_F^2 \\
&\leq \left\| \boldsymbol{W} - \frac{1}{n}\mathbf{1}\mathbf{1}^\top \right\|_{\mathrm{op}}^2 \left\| \boldsymbol{X} - \bar{\boldsymbol{X}} \right\|_F^2 \\
&\leq \max_{i \geq 2}(\lambda_i^2) \left\| \boldsymbol{X} - \bar{\boldsymbol{X}} \right\|_F^2,
\end{aligned}$$

where we use Lemma 5 and the assumption that $\boldsymbol{W}$ is symmetric in the last inequality. Moreover, from Lemma 5, it holds that $\max_{i \geq 2}(\lambda_i^2) < 1$, and we can conclude the statement. $\square$

**Lemma 6.** *Let $X_1, X_2, \ldots, X_n$ be d-dimensional independent random variables with the same mean, and assume that there exists $\sigma$ such that $\mathbb{E}\|X_i - \mathbb{E}[X_i]\|^2 \leq \sigma^2$ for all $i$. Define $X$ as follows:*

$$X := (X_1, X_2, \ldots, X_n),$$

*where $X$ is random variable of dimension $d \times n$. Then, it holds that*

$$\frac{1}{n}\mathbb{E} \left\| X \left( \boldsymbol{W} - \frac{1}{n}\mathbf{1}\mathbf{1}^\top \right) \right\|_F^2 \leq \frac{\sigma^2}{n} \sum_{i=2}^n \lambda_i^2.$$

*Proof.* We have

$$\begin{aligned}
\mathbb{E} \left\| X \left( \boldsymbol{W} - \frac{1}{n}\mathbf{1}\mathbf{1}^\top \right) \right\|_F^2 &= \mathbb{E} \left\| (X - \mathbb{E}[X]) \left( \boldsymbol{W} - \frac{1}{n}\mathbf{1}\mathbf{1}^\top \right) \right\|_F^2 \\
&= \mathbb{E}\,\mathrm{Tr} \left( \left( \boldsymbol{W} - \frac{1}{n}\mathbf{1}\mathbf{1}^\top \right) (X - \mathbb{E}[X])^\top (X - \mathbb{E}[X]) \left( \boldsymbol{W} - \frac{1}{n}\mathbf{1}\mathbf{1}^\top \right) \right) \\
&= \mathrm{Tr} \left( \left( \boldsymbol{W} - \frac{1}{n}\mathbf{1}\mathbf{1}^\top \right) \mathbb{E} \left[ (X - \mathbb{E}[X])^\top (X - \mathbb{E}[X]) \right] \left( \boldsymbol{W} - \frac{1}{n}\mathbf{1}\mathbf{1}^\top \right) \right),
\end{aligned}$$

where we use the assumption that $X_1, \ldots, X_n$ has the same mean in the first equality. Since $X_1 - \mathbb{E}[X_1], \ldots, X_n - \mathbb{E}[X_n]$ are independent and have the mean of $\mathbf{0}$, we have

$$\mathbb{E} \left[ (X - \mathbb{E}[X])^\top (X - \mathbb{E}[X]) \right] = \begin{pmatrix} \sigma_1^2 & & & \\ & \sigma_2^2 & & \\ & & \ddots & \\ & & & \sigma_n^2 \end{pmatrix},$$

where $\sigma_i^2 := \mathbb{E}\|X_i - \mathbb{E}[X_i]\|^2$. Thus, we obtain

$$
\mathbb{E}\left\|X\left(\boldsymbol{W} - \frac{1}{n}\mathbf{1}\mathbf{1}^\top\right)\right\|_F^2 = \mathrm{Tr}\left(\left(\boldsymbol{W} - \frac{1}{n}\mathbf{1}\mathbf{1}^\top\right)\begin{pmatrix}\sigma_1^2 & & & \\ & \sigma_2^2 & & \\ & & \ddots & \\ & & & \sigma_n^2\end{pmatrix}\left(\boldsymbol{W} - \frac{1}{n}\mathbf{1}\mathbf{1}^\top\right)\right)
$$

$$
= \left\|\begin{pmatrix}\sigma_1 & & & \\ & \sigma_2 & & \\ & & \ddots & \\ & & & \sigma_n\end{pmatrix}\left(\boldsymbol{W} - \frac{1}{n}\mathbf{1}\mathbf{1}^\top\right)\right\|_F^2
$$

$$
\leq \left\|\begin{pmatrix}\sigma_1 & & & \\ & \sigma_2 & & \\ & & \ddots & \\ & & & \sigma_n\end{pmatrix}\right\|_{op}^2 \left\|\boldsymbol{W} - \frac{1}{n}\mathbf{1}\mathbf{1}^\top\right\|_F^2
$$

$$
\leq \sigma^2 \left\|\boldsymbol{W} - \frac{1}{n}\mathbf{1}\mathbf{1}^\top\right\|_F^2,
$$

where we use $\sigma_i^2 \leq \sigma^2$ in the last inequality. From Lemma 5, $(\boldsymbol{W} - \frac{1}{n}\mathbf{1}\mathbf{1}^\top)^2$ has eigenvalues $0, \lambda_2^2, \lambda_3^2, \ldots, \lambda_n^2$. Thus, we obtain

$$
\mathbb{E}\left\|X\left(\boldsymbol{W} - \frac{1}{n}\mathbf{1}\mathbf{1}^\top\right)\right\|_F^2 \leq \sigma^2 \sum_{i=2}^n \lambda_i^2.
$$

Dividing both sides by $n$, we obtain the desired statement. $\qquad\square$

### C.4 PROOF IN CONVEX CASE

**Lemma 7.** *Suppose that Assumptions 1, 2 and 5 to 7 hold. When $\eta \leq \frac{1}{12L}$, it holds that*

$$\mathbb{E}\left\|\bar{\boldsymbol{x}}^{(r+1)} - \boldsymbol{x}^\star\right\|^2 \leq \left(1 - \frac{\mu\eta}{2}\right)\mathbb{E}\left\|\bar{\boldsymbol{x}}^{(r)} - \boldsymbol{x}^\star\right\|^2 + \frac{\eta^2\sigma_\star^2}{n} - \eta\left(\mathbb{E}f(\bar{\boldsymbol{x}}^{(r)}) - f^\star\right) + 3L\eta\Xi^{(r,0)}.$$

*Proof.* See Lemma 8 in Koloskova et al. (2020b). □

**Lemma 8.** *Suppose that Assumptions 1, 2 and 5 to 7 hold. When $\eta \leq \frac{p}{5L}$, it holds that*

$$\Xi^{(r+1,k)} \leq \left(1 + \frac{p}{2}\right)\Xi^{(r,k+1)} + \frac{p}{4}(1-p)^{k+1}\Xi^{(r,0)}$$

$$+ \frac{58L}{p}(1-p)^{k+1}\eta^2\left(f(\bar{\boldsymbol{x}}^{(r)}) - f^\star\right) + \frac{5\sigma_\star^2\eta^2}{n}\sum_{i=2}^n \lambda_i^{2(k+1)} + \frac{9\zeta_\star^2\eta^2}{p}(1-p)^{k+1}. \quad (9)$$

*where*

$$\Xi^{(r+1,k)} := \frac{1}{n}\mathbb{E}\left\|\boldsymbol{X}^{(r+1)}\boldsymbol{W}^k - \bar{\boldsymbol{X}}^{(r+1)}\right\|^2. \quad (10)$$

*Proof.* We have

$$\mathbb{E}_r\left\|\boldsymbol{X}^{(r+1)}\boldsymbol{W}^k - \bar{\boldsymbol{X}}^{(r+1)}\right\|_F^2$$

$$= \mathbb{E}_r\left\|\boldsymbol{X}^{(r)}\boldsymbol{W}^{k+1} - \bar{\boldsymbol{X}}^{(r)} - \eta\nabla F(\boldsymbol{X}^{(r)};\xi^{(r)})\boldsymbol{W}^{k+1} + \eta\nabla F(\boldsymbol{X}^{(r)};\xi^{(r)})\frac{1}{n}\mathbf{1}\mathbf{1}^\top\right\|_F^2$$

$$= \underbrace{\left\|\boldsymbol{X}^{(r)}\boldsymbol{W}^{k+1} - \bar{\boldsymbol{X}}^{(r)} - \eta\nabla F(\boldsymbol{X}^{(r)})\boldsymbol{W}^{k+1} + \eta\nabla F(\boldsymbol{X}^{(r)})\frac{1}{n}\mathbf{1}\mathbf{1}^\top\right\|_F^2}_{T_1}$$

$$+ \eta^2\underbrace{\mathbb{E}_r\left\|\left(\nabla F(\boldsymbol{X}^{(r)};\xi^{(r)}) - \nabla F(\boldsymbol{X}^{(r)})\right)\left(\boldsymbol{W}^{k+1} - \frac{1}{n}\mathbf{1}\mathbf{1}^\top\right)\right\|_F^2}_{T_2}.$$

$T_1$ and $T_2$ are bounded as follows:

$$T_1 \leq \left(1 + \frac{p}{2}\right)\left\|\boldsymbol{X}^{(r)}\boldsymbol{W}^{k+1} - \bar{\boldsymbol{X}}^{(r)}\right\|_F^2 + \left(1 + \frac{2}{p}\right)\eta^2\left\|\nabla F(\boldsymbol{X}^{(r)})\left(\boldsymbol{W}^{k+1} - \frac{1}{n}\mathbf{1}\mathbf{1}^\top\right)\right\|_F^2$$

$$\leq \left(1 + \frac{p}{2}\right)\left\|\boldsymbol{X}^{(r)}\boldsymbol{W}^{k+1} - \bar{\boldsymbol{X}}^{(r)}\right\|_F^2$$

$$+ \frac{3}{p}\eta^2\left\|\left(\nabla F(\boldsymbol{X}^{(r)}) \pm \nabla F(\bar{\boldsymbol{X}}^{(r)}) \pm \nabla F(\boldsymbol{X}^\star)\right)\left(\boldsymbol{W}^{k+1} - \frac{1}{n}\mathbf{1}\mathbf{1}^\top\right)\right\|_F^2$$

$$\leq \left(1 + \frac{p}{2}\right)\left\|\boldsymbol{X}^{(r)}\boldsymbol{W}^{k+1} - \bar{\boldsymbol{X}}^{(r)}\right\|_F^2$$

$$+ \frac{3}{p}\eta^2\left\|\nabla F(\boldsymbol{X}^{(r)}) \pm \nabla F(\bar{\boldsymbol{X}}^{(r)}) \pm \nabla F(\boldsymbol{X}^\star)\right\|_F^2\left\|\boldsymbol{W}^{k+1} - \frac{1}{n}\mathbf{1}\mathbf{1}^\top\right\|_{\text{op}}^2$$

$$\leq \left(1 + \frac{p}{2}\right)\left\|\boldsymbol{X}^{(r)}\boldsymbol{W}^{k+1} - \bar{\boldsymbol{X}}^{(r)}\right\|_F^2 + \frac{3}{p}(1-p)^{k+1}\eta^2\left\|\nabla F(\boldsymbol{X}^{(r)}) \pm \nabla F(\bar{\boldsymbol{X}}^{(r)}) \pm \nabla F(\boldsymbol{X}^\star)\right\|_F^2$$

$$\leq \left(1 + \frac{p}{2}\right)\left\|\boldsymbol{X}^{(r)}\boldsymbol{W}^{k+1} - \bar{\boldsymbol{X}}^{(r)}\right\|_F^2 + \frac{9}{p}(1-p)^{k+1}\eta^2\left\|\nabla F(\boldsymbol{X}^{(r)}) - \nabla F(\bar{\boldsymbol{X}}^{(r)})\right\|_F^2$$

$$+ \frac{9}{p}(1-p)^{k+1}\eta^2\left\|\nabla F(\bar{\boldsymbol{X}}^{(r)}) - \nabla F(\boldsymbol{X}^\star)\right\|_F^2 + \frac{9}{p}(1-p)^{k+1}\eta^2\left\|\nabla F(\boldsymbol{X}^\star)\right\|_F^2$$

$$\leq \left(1 + \frac{p}{2}\right)\left\|\boldsymbol{X}^{(r)}\boldsymbol{W}^{k+1} - \bar{\boldsymbol{X}}^{(r)}\right\|_F^2 + \frac{9L^2}{p}(1-p)^{k+1}\eta^2\left\|\boldsymbol{X}^{(r)} - \bar{\boldsymbol{X}}^{(r)}\right\|_F^2$$

$$+ \frac{18Ln}{p}(1-p)^{k+1}\eta^2\left(f(\bar{\boldsymbol{x}}^{(r)}) - f^\star\right) + \frac{9n\zeta_\star^2}{p}(1-p)^{k+1}\eta^2,$$

where we use $\|\boldsymbol{W}^{k+1} - \frac{1}{n}\mathbf{1}\mathbf{1}^\top\|_{\mathrm{op}}^2 = (1-p)^{k+1}$.

$$T_2 \leq 5\mathbb{E}_r \left\| \left( \nabla F(\boldsymbol{X}^{(r)}; \xi^{(r)}) - \nabla F(\bar{\boldsymbol{X}}^{(r)}; \xi^{(r)}) \right) \left( \boldsymbol{W}^{k+1} - \frac{1}{n}\mathbf{1}\mathbf{1}^\top \right) \right\|_F^2$$

$$+ 5\mathbb{E}_r \left\| \left( \nabla F(\bar{\boldsymbol{X}}^{(r)}; \xi^{(r)}) - \nabla F(\boldsymbol{X}^\star; \xi^{(r)}) \right) \left( \boldsymbol{W}^{k+1} - \frac{1}{n}\mathbf{1}\mathbf{1}^\top \right) \right\|_F^2$$

$$+ 5\mathbb{E}_r \left\| \left( \nabla F(\boldsymbol{X}^\star; \xi^{(r)}) - \nabla F(\boldsymbol{X}^\star) \right) \left( \boldsymbol{W}^{k+1} - \frac{1}{n}\mathbf{1}\mathbf{1}^\top \right) \right\|_F^2$$

$$+ 5\mathbb{E}_r \left\| \left( \nabla F(\boldsymbol{X}^\star) - \nabla F(\bar{\boldsymbol{X}}^{(r)}) \right) \left( \boldsymbol{W}^{k+1} - \frac{1}{n}\mathbf{1}\mathbf{1}^\top \right) \right\|_F^2$$

$$+ 5\mathbb{E}_r \left\| \left( \nabla F(\bar{\boldsymbol{X}}^{(r)}) - \nabla F(\boldsymbol{X}^{(r)}) \right) \left( \boldsymbol{W}^{k+1} - \frac{1}{n}\mathbf{1}\mathbf{1}^\top \right) \right\|_F^2$$

$$\leq 10(1-p)^{k+1}L^2 \left\| \boldsymbol{X}^{(r)} - \bar{\boldsymbol{X}}^{(r)} \right\|_F^2 + 40(1-p)^{k+1}n \left( f(\bar{\boldsymbol{x}}^{(r)}) - f^\star \right)$$

$$+ 5\mathbb{E}_r \left\| \left( \nabla F(\boldsymbol{X}^\star; \xi^{(r)}) - \nabla F(\boldsymbol{X}^\star) \right) \left( \boldsymbol{W}^{k+1} - \frac{1}{n}\mathbf{1}\mathbf{1}^\top \right) \right\|_F^2.$$

Using Lemma 6, we have

$$T_2 \leq 10(1-p)^{k+1}L^2 \left\| \boldsymbol{X}^{(r)} - \bar{\boldsymbol{X}}^{(r)} \right\|_F^2 + 40L(1-p)^{k+1}n \left( f(\bar{\boldsymbol{x}}^{(r)}) - f^\star \right) + 5\sigma_\star^2 \sum_{i=2}^n \lambda_i^{2(k+1)}.$$

Combining the above two inequalities, we have

$$\Xi^{(r+1,k)} \leq \left( 1 + \frac{p}{2} \right) \Xi^{(r,k+1)} + \frac{19L^2}{p}(1-p)^{k+1}\eta^2 \Xi^{(r,0)}$$

$$+ \frac{49}{p}(1-p)^{k+1}\eta^2 \left( f(\bar{\boldsymbol{x}}^{(r)}) - f^\star \right) + \frac{5\sigma_\star^2\eta^2}{n} \sum_{i=2}^n \lambda_i^{2(k+1)} + \frac{9\zeta_\star^2\eta^2}{p}(1-p)^{k+1}.$$

Using $\eta \leq \frac{p}{5L}$, we obtain the desired result. $\qquad\square$

**Lemma 9.** *Suppose that Assumptions 1, 2 and 5 to 7 hold, and $\{\boldsymbol{x}_i^{(0)}\}_{i=1}^n$ are initialized to the same value. When $\eta \leq \frac{p}{480L}$, it holds that*

$$\Xi^{(r,k)} \leq C(r,k) + (1-p)^{k+1}\frac{p}{4} \sum_{r'=1}^{r-1} \left( (1-p)(1+\frac{3p}{4}) \right)^{r-r'-1} C(r',0)$$

$$+ \frac{58L\eta^2}{p}(1-p)^{k+1} \sum_{r'=0}^{r-1} \left( (1-p)(1+\frac{3p}{4}) \right)^{r-r'-1} \left( \mathbb{E}f(\bar{\boldsymbol{x}}^{(r')}) - f^\star \right)$$

$$+ \frac{9\zeta_\star^2\eta^2}{p}(1-p)^{k+1} \sum_{r'=0}^{r-1} \left( (1-p)(1+\frac{3p}{4}) \right)^{r-r'-1} \tag{11}$$

*where*

$$C(r,k) := \frac{5\sigma_\star^2\eta^2}{n} \sum_{i=2}^n \lambda_i^{2(k+1)} \sum_{r'=0}^{r-1} \left( \lambda_i^2(1+\frac{p}{2}) \right)^{r'}.$$

*Proof.* When $r = 1$, Eq. (11) holds since $\Xi^{(0,k)} = 0$ for any $k$.

Suppose that Eq. (11) holds when $r = r''$. We have

$$\Xi^{(r''+1,k)} \leq \left(1 + \frac{p}{2}\right) \Xi^{(r'',k+1)} + \frac{p}{4}(1-p)^{k+1} \Xi^{(r'',0)}$$

$$+ \frac{58L}{p}(1-p)^{k+1}\eta^2 \left(f(\bar{\boldsymbol{x}}^{(r'')}) - f^\star\right) + \frac{5\sigma_\star^2\eta^2}{n} \sum_{i=2}^{n} \lambda_i^{2(k+1)} + \frac{9\zeta_\star^2\eta^2}{p}(1-p)^{k+1}$$

$$\leq \left(1 + \frac{p}{2}\right) C(r'', k+1) + \frac{5\sigma_\star^2\eta^2}{n} \sum_{i=2}^{n} \lambda_i^{2(k+1)}$$

$$+ (1-p)^{k+1}\frac{p}{4} \sum_{r'=1}^{r''} \left((1-p)(1+\frac{3p}{4})\right)^{r''-r'} C(r', 0)$$

$$+ \frac{58L\eta^2}{p}(1-p)^{k+1} \sum_{r'=0}^{r''} \left((1-p)(1+\frac{3p}{4})\right)^{r''-r'} \left(\mathbb{E}f(\bar{\boldsymbol{x}}^{(r')}) - f^\star\right)$$

$$+ \frac{9\zeta_\star^2\eta^2}{p}(1-p)^{k+1} \sum_{r'=0}^{r''} \left((1-p)(1+\frac{3p}{4})\right)^{r''-r'}.$$

Using

$$\left(1 + \frac{p}{2}\right) C(r'', k+1) + \frac{5\sigma_\star^2\eta^2}{n} \sum_{i=2}^{n} \lambda_i^{2(k+1)} = C(r''+1, k),$$

Eq. (11) holds when $r = r'' + 1$. Using mathematical induction, we can conclude the statement. $\square$

**Lemma 10.** *Suppose that Assumptions 1, 2 and 5 to 7 hold, and $\{\boldsymbol{x}_i^{(0)}\}_{i=1}^{n}$ are initialized to the same value. When $\eta \leq \frac{p}{5L}$, it holds that*

$$\Xi^{(r,0)} \leq \frac{10\sigma_\star^2\eta^2}{n} \sum_{i=2}^{n} \frac{\lambda_i^2}{1-\lambda_i^2} + \frac{36(1-p)\zeta_\star^2\eta^2}{p^2}$$

$$+ \frac{58L\eta^2}{p}(1-p) \sum_{r'=0}^{r-1} \left((1-p)(1+\frac{3p}{4})\right)^{r-r'-1} \left(\mathbb{E}f(\bar{\boldsymbol{x}}^{(r')}) - f^\star\right). \qquad (12)$$

*Proof.* From Lemma 9, we have

$$\Xi^{(r,0)} \leq C(r, 0) + \underbrace{(1-p)\frac{p}{4} \sum_{r'=1}^{r-1} \left((1-p)(1+\frac{3p}{4})\right)^{r-r'-1} C(r', 0)}_{T_1}$$

$$+ \frac{58L\eta^2}{p}(1-p) \sum_{r'=0}^{r-1} \left((1-p)(1+\frac{3p}{4})\right)^{r-r'-1} \left(\mathbb{E}f(\bar{\boldsymbol{x}}^{(r')}) - f^\star\right)$$

$$+ \underbrace{\frac{9\zeta_\star^2\eta^2}{p}(1-p) \sum_{r'=0}^{r-1} \left((1-p)(1+\frac{3p}{4})\right)^{r-r'-1}}_{T_2}.$$

We have

$$C(r, 0) = \frac{5\sigma_\star^2\eta^2}{n} \sum_{i=2}^{n} \lambda_i^2 \sum_{r'=0}^{r-1} \left(\lambda_i^2(1+\frac{p}{2})\right)^{r'}$$

$$\leq \frac{5\sigma_\star^2\eta^2}{n} \sum_{i=2}^{n} \lambda_i^2 \sum_{r'=0}^{r-1} \left(1 - \frac{1-\lambda_i^2}{2}\right)^{r'}$$

$$\leq \frac{5\sigma_\star^2\eta^2}{n} \sum_{i=2}^{n} \frac{\lambda_i^2}{1-\lambda_i^2},$$

where we use $p = 1 - \max_{i \geq 2}(\lambda_i^2) \leq 1 - \lambda_i^2$ for all $i \geq 2$ in the first inequality.

$T_1$ is bounded as follows:

$$
\begin{aligned}
T_1 &= (1-p)\frac{5\sigma_\star^2\eta^2 p}{4n} \sum_{i=2}^{n} \lambda_i^2 \sum_{r'=1}^{r-1} \sum_{r''=0}^{r'-1} \left(\lambda_i^2(1+\frac{p}{2})\right)^{r''} \left((1-p)(1+\frac{3p}{4})\right)^{r-r'-1} \\
&= (1-p)\frac{5\sigma_\star^2\eta^2 p}{4n} \sum_{i=2}^{n} \lambda_i^2 \sum_{r''=0}^{r-2} \left(\lambda_i^2(1+\frac{p}{2})\right)^{r''} \sum_{r'=r''+1}^{r-1} \left((1-p)(1+\frac{3p}{4})\right)^{r-r'-1} \\
&\leq (1-p)\frac{5\sigma_\star^2\eta^2 p}{4n} \sum_{i=2}^{n} \lambda_i^2 \sum_{r''=0}^{r-2} \left(\lambda_i^2(1+\frac{p}{2})\right)^{r''} \sum_{r'=r''+1}^{r-1} \left(1-\frac{p}{4}\right)^{r-r'-1} \\
&\leq (1-p)\frac{5\sigma_\star^2\eta^2}{n} \sum_{i=2}^{n} \lambda_i^2 \sum_{r''=0}^{r-2} \left(\lambda_i^2(1+\frac{p}{2})\right)^{r''} \\
&\leq (1-p)\frac{5\sigma_\star^2\eta^2}{n} \sum_{i=2}^{n} \lambda_i^2 \sum_{r''=0}^{r-2} \left(1 - \frac{1-\lambda_i^2}{2}\right)^{r''} \\
&\leq (1-p)\frac{5\sigma_\star^2\eta^2}{n} \sum_{i=2}^{n} \frac{\lambda_i^2}{1-\lambda_i^2}.
\end{aligned}
$$

$T_2$ is bounded as follows:

$$
T_2 \leq \frac{9\zeta_\star^2\eta^2}{p}(1-p) \sum_{r'=0}^{r-1} \left(1-\frac{p}{4}\right)^{r-r'-1} \leq \frac{36(1-p)\zeta_\star^2\eta^2}{p^2}
$$

Combining the above two inequalities, we obtain the desired result. $\qquad\square$

**Lemma 11.** *Suppose that Assumptions 1, 2 and 5 to 7 hold, and $\{\boldsymbol{x}_i^{(0)}\}_{i=1}^{n}$ are initialized to the same value. When $\eta \leq \frac{p}{5L}$, it holds that*

$$
\frac{1}{R+1} \sum_{r=0}^{R} \Xi^{(r,0)} \leq \frac{10\sigma_\star^2\eta^2}{n} \sum_{i=2}^{n} \frac{\lambda_i^2}{1-\lambda_i^2} + \frac{36(1-p)\zeta_\star^2\eta^2}{p^2} + \frac{240L\eta^2}{p^2} \frac{(1-p)}{R+1} \sum_{r'=0}^{R-1} \left(\mathbb{E}f(\bar{\boldsymbol{x}}^{(r')}) - f^\star\right).
$$

*Proof.* Using $\Xi^{(0,0)} = 0$ and Lemma 10, we have

$$
\begin{aligned}
\frac{1}{R+1} \sum_{r=0}^{R} \Xi^{(r,0)} &\leq \frac{10\sigma_\star^2\eta^2}{n} \sum_{i=2}^{n} \frac{\lambda_i^2}{1-\lambda_i^2} + \frac{36(1-p)\zeta_\star^2\eta^2}{p^2} \\
&\quad + \frac{58L\eta^2}{p} \frac{(1-p)}{R+1} \sum_{r=1}^{R} \sum_{r'=0}^{r-1} \left((1-p)(1+\frac{3p}{4})\right)^{r-r'-1} \left(\mathbb{E}f(\bar{\boldsymbol{x}}^{(r')}) - f^\star\right) \\
&= \frac{10\sigma_\star^2\eta^2}{n} \sum_{i=2}^{n} \frac{\lambda_i^2}{1-\lambda_i^2} + \frac{36(1-p)\zeta_\star^2\eta^2}{p^2} \\
&\quad + \frac{58\eta^2}{p} \frac{(1-p)}{R+1} \sum_{r'=0}^{R-1} \left(\mathbb{E}f(\bar{\boldsymbol{x}}^{(r')}) - f^\star\right) \sum_{r=r'+1}^{R} \left((1-p)(1+\frac{3p}{4})\right)^{r-r'-1} \\
&\leq \frac{10\sigma_\star^2\eta^2}{n} \sum_{i=2}^{n} \frac{\lambda_i^2}{1-\lambda_i^2} + \frac{36(1-p)\zeta_\star^2\eta^2}{p^2} \\
&\quad + \frac{58L\eta^2}{p} \frac{(1-p)}{R+1} \sum_{r'=0}^{R-1} \left(\mathbb{E}f(\bar{\boldsymbol{x}}^{(r')}) - f^\star\right) \sum_{r=r'+1}^{R} \left(1-\frac{p}{4}\right)^{r-r'-1} \\
&\leq \frac{10\sigma_\star^2\eta^2}{n} \sum_{i=2}^{n} \frac{\lambda_i^2}{1-\lambda_i^2} + \frac{36(1-p)\zeta_\star^2\eta^2}{p^2} + \frac{240L\eta^2}{p^2} \frac{(1-p)}{R+1} \sum_{r'=0}^{R-1} \left(\mathbb{E}f(\bar{\boldsymbol{x}}^{(r')}) - f^\star\right).
\end{aligned}
$$

$\square$

**Lemma 12.** *Suppose that Assumptions 1, 2 and 5 to 7 hold, and $\{\boldsymbol{x}_i^{(0)}\}_{i=1}^n$ are initialized to the same value. When $\eta \le \frac{p}{5L}$, it holds that*

$$\frac{1}{W_R} \sum_{r=0}^R w_r \Xi^{(r,0)} \le \frac{10\sigma_\star^2 \eta^2}{n} \sum_{i=2}^n \frac{\lambda_i^2}{1-\lambda_i^2} + \frac{36(1-p)\zeta_\star^2 \eta^2}{p^2}$$

$$+ \frac{3712L\eta^2}{7p^2} \frac{(1-p)}{R+1} \sum_{r'=0}^{R-1} w_{r'} \left( \mathbb{E}f(\bar{\boldsymbol{x}}^{(r')}) - f^\star \right),$$

*where $w_r$ satisfies $0 < w_{r+1} \le w_r \le (1 + \frac{p}{32})w_{r+1}$ and $W_R := \sum_{r=0}^R w_r$.*

*Proof.* Using $\Xi^{(0,0)} = 0$ and Lemma 10, we have

$$\frac{1}{W_R} \sum_{r=0}^R \Xi^{(r,0)} \le \frac{10\sigma_\star^2 \eta^2}{n} \sum_{i=2}^n \frac{\lambda_i^2}{1-\lambda_i^2} + \frac{36(1-p)\zeta_\star^2 \eta^2}{p^2}$$

$$+ \frac{58L\eta^2}{p} \frac{(1-p)}{W_R} \sum_{r=1}^R w_r \sum_{r'=0}^{r-1} \left( (1-p)(1+\frac{3p}{4}) \right)^{r-r'-1} \left( \mathbb{E}f(\bar{\boldsymbol{x}}^{(r')}) - f^\star \right)$$

$$\le \frac{10\sigma_\star^2 \eta^2}{n} \sum_{i=2}^n \frac{\lambda_i^2}{1-\lambda_i^2} + \frac{36(1-p)\zeta_\star^2 \eta^2}{p^2}$$

$$+ \frac{116L\eta^2}{p} \frac{(1-p)}{W_R} \sum_{r=1}^R \sum_{r'=0}^{r-1} \left( 1+\frac{p}{32} \right)^{r-r'-1} \left( 1-\frac{p}{4} \right)^{r-r'-1} w_{r'} \left( \mathbb{E}f(\bar{\boldsymbol{x}}^{(r')}) - f^\star \right)$$

$$\le \frac{10\sigma_\star^2 \eta^2}{n} \sum_{i=2}^n \frac{\lambda_i^2}{1-\lambda_i^2} + \frac{36(1-p)\zeta_\star^2 \eta^2}{p^2}$$

$$+ \frac{116\eta^2}{p} \frac{(1-p)}{W_R} \sum_{r'=0}^{R-1} w_{r'} \left( \mathbb{E}f(\bar{\boldsymbol{x}}^{(r')}) - f^\star \right) \sum_{r=r'+1}^R \left( 1-\frac{7p}{32} \right)^{r-r'-1}$$

$$\le \frac{10\sigma_\star^2 \eta^2}{n} \sum_{i=2}^n \frac{\lambda_i^2}{1-\lambda_i^2} + \frac{36(1-p)\zeta_\star^2 \eta^2}{p^2}$$

$$+ \frac{3712L\eta^2}{7p^2} \frac{(1-p)}{W_R} \sum_{r'=0}^{R-1} w_{r'} \left( \mathbb{E}f(\bar{\boldsymbol{x}}^{(r')}) - f^\star \right).$$

$\square$

**Lemma 13.** *Suppose that Assumptions 1, 2 and 5 to 7 hold with $\mu = 0$, and $\{\boldsymbol{x}_i^{(0)}\}_{i=1}^n$ are initialized to the same value. There exists a stepsize $\eta \le \frac{p}{48L}$ such that*

$$\frac{1}{R+1} \sum_{r=0}^R (\mathbb{E}f(\bar{\boldsymbol{x}}^{(r)}) - f^\star) \le \mathcal{O}\left( \sqrt{\frac{r_0\sigma_\star^2}{nR}} + \left( \left( \frac{\sigma_\star^2}{n} \sum_{i=2}^n \frac{\lambda_i^2}{1-\lambda_i^2} + \frac{(1-p)\zeta_\star^2}{p^2} \right) \frac{Lr_0^2}{R^2} \right)^{\frac{1}{3}} + \frac{Lr_0}{Rp} \right),$$

*where $r_0 := \|\bar{\boldsymbol{x}}^{(0)} - \boldsymbol{x}^\star\|^2$.*

*Proof.* From Lemma 7, we have

$$\frac{1}{R+1} \sum_{r=0}^R \left( \mathbb{E}f(\bar{\boldsymbol{x}}^{(r)}) - f^\star \right) \le \frac{\left\| \bar{\boldsymbol{x}}^{(0)} - \boldsymbol{x}^\star \right\|^2}{\eta R} + \frac{\eta\sigma_\star^2}{n} + \frac{3L}{R+1} \sum_{r=0}^R \Xi^{(r,0)}.$$

Using Lemma 10, we obtain

$$\frac{1}{R+1} \sum_{r=0}^{R} \left( \mathbb{E} f(\bar{\boldsymbol{x}}^{(r)}) - f^{\star} \right)$$

$$\leq \frac{\left\| \bar{\boldsymbol{x}}^{(0)} - \boldsymbol{x}^{\star} \right\|^2}{\eta R} + \frac{\eta \sigma_{\star}^2}{n} + \frac{30 L \sigma_{\star}^2 \eta^2}{n} \sum_{i=2}^{n} \frac{\lambda_i^2}{1 - \lambda_i^2} + \frac{108 L (1-p) \zeta_{\star}^2 \eta^2}{p^2} + \frac{720 L^2 \eta^2}{p^2} \frac{(1-p)}{R+1} \sum_{r'=0}^{R-1} \left( \mathbb{E} f(\bar{\boldsymbol{x}}^{(r')}) - f^{\star} \right).$$

Using $\eta \leq \frac{p}{48L}$, we have

$$\frac{1}{2(R+1)} \sum_{r=0}^{R} \left( \mathbb{E} f(\bar{\boldsymbol{x}}^{(r)}) - f^{\star} \right)$$

$$\leq \frac{\left\| \bar{\boldsymbol{x}}^{(0)} - \boldsymbol{x}^{\star} \right\|^2}{\eta R} + \frac{\eta \sigma_{\star}^2}{n} + \frac{30 L \sigma_{\star}^2 \eta^2}{n} \sum_{i=2}^{n} \frac{\lambda_i^2}{1 - \lambda_i^2} + \frac{108 L (1-p) \zeta_{\star}^2 \eta^2}{p^2}.$$

Using Lemma 17 in Koloskova et al. (2020b) and $\eta \leq \frac{p}{48L}$, we can conclude the statement. $\qquad \square$

**Lemma 14.** *Suppose that Assumptions 1, 2 and 5 to 7 hold with $\mu > 0$, and $\{\boldsymbol{x}_i^{(0)}\}_{i=1}^{n}$ are initialized to the same value. There exists a stepsize $\eta \leq \frac{p}{60L}$ such that $\frac{1}{W_R} \sum_{r=0}^{r} (\mathbb{E} f(\bar{\boldsymbol{x}}^{(r)}) - f^{\star})$ is bounded from above by*

$$\tilde{\mathcal{O}} \left( \frac{\sigma^2}{n \mu R} + \left( \frac{\sigma_{\star}^2}{n} \sum_{i=2}^{n} \frac{\lambda_i^2}{1 - \lambda_i^2} + \frac{(1-p) \zeta_{\star}^2}{p^2} \right) \frac{L}{R^2} + \frac{L r_0}{p} \exp \left[ -\frac{\mu p R}{L} \right] \right),$$

*where $w_r := (1 - \frac{\mu \eta}{2})^{-(r+1)}$, $W_R := \sum_{r=0}^{R} w_r$, $\bar{\boldsymbol{x}}^{(r)} := \frac{1}{n} \sum_{i=1}^{n} \boldsymbol{x}_i^{(r)}$ and $r_0 := \| \bar{\boldsymbol{x}}^{(0)} - \boldsymbol{x}^{\star} \|^2$.*

*Proof.* From Lemma 7, we have

$$\frac{1}{W_R} \sum_{r=0}^{R} w_r \left( \mathbb{E} f(\bar{\boldsymbol{x}}^{(r)}) - f^{\star} \right) \leq \frac{1}{W_R} \sum_{r=0}^{R} \left( \left(1 - \frac{\mu \eta}{2}\right) w_r \frac{\left\| \bar{\boldsymbol{x}}^{(r)} - \boldsymbol{x}^{\star} \right\|^2}{\eta} - w_r \frac{\left\| \bar{\boldsymbol{x}}^{(r+1)} - \boldsymbol{x}^{\star} \right\|^2}{\eta} \right)$$

$$+ \frac{\eta \sigma_{\star}^2}{n} + \frac{3L}{W_R} \sum_{r=0}^{R} w_r \Xi^{(r,0)}.$$

Using Lemma Lemma 12, we obtain

$$\frac{1}{W_R} \sum_{r=0}^{R} w_r \left( \mathbb{E} f(\bar{\boldsymbol{x}}^{(r)}) - f^{\star} \right) \leq \frac{1}{W_R} \sum_{r=0}^{R} \left( \left(1 - \frac{\mu \eta}{2}\right) w_r \frac{\left\| \bar{\boldsymbol{x}}^{(r)} - \boldsymbol{x}^{\star} \right\|^2}{\eta} - w_r \frac{\left\| \bar{\boldsymbol{x}}^{(r+1)} - \boldsymbol{x}^{\star} \right\|^2}{\eta} \right)$$

$$+ \frac{32 L \sigma_{\star}^2 \eta^2}{n} \sum_{i=2} \frac{\lambda_i^2}{1 - \lambda_i^2} + \frac{108 L (1-p)}{p^2} \zeta_{\star}^2 \eta^2$$

$$+ \frac{11136 L^2 (1-p)}{7 p^2 W_R} \eta^2 \sum_{r=0}^{R-1} w_r \left( \mathbb{E} f(\bar{\boldsymbol{x}}^{(r)}) - f^{\star} \right).$$

Using $\eta \leq \frac{p}{60L}$, we have

$$\frac{1}{2W_R} \sum_{r=0}^{R} w_r \left( \mathbb{E} f(\bar{\boldsymbol{x}}^{(r)}) - f^{\star} \right) \leq \frac{1}{W_R} \sum_{r=0}^{R} \left( \left(1 - \frac{\mu \eta}{2}\right) w_r \frac{\left\| \bar{\boldsymbol{x}}^{(r)} - \boldsymbol{x}^{\star} \right\|^2}{\eta} - w_r \frac{\left\| \bar{\boldsymbol{x}}^{(r+1)} - \boldsymbol{x}^{\star} \right\|^2}{\eta} \right)$$

$$+ \frac{32 L \sigma_{\star}^2 \eta^2}{n} \sum_{i=2} \frac{\lambda_i^2}{1 - \lambda_i^2} + \frac{108 L (1-p)}{p^2} \zeta_{\star}^2 \eta^2.$$

Using the definition of $w_r$, we get

$$\frac{1}{2W_R} \sum_{r=0}^{R} w_r \left( \mathbb{E} f(\bar{\boldsymbol{x}}^{(r)}) - f^{\star} \right) \leq \frac{1}{W_R} \left( \left(1 - \frac{\mu \eta}{2}\right) w_0 \frac{\left\| \bar{\boldsymbol{x}}^{(0)} - \boldsymbol{x}^{\star} \right\|^2}{\eta} - w_R \frac{\left\| \bar{\boldsymbol{x}}^{(R+1)} - \boldsymbol{x}^{\star} \right\|^2}{\eta} \right)$$

$$+ \frac{32 L \sigma_{\star}^2 \eta^2}{n} \sum_{i=2} \frac{\lambda_i^2}{1 - \lambda_i^2} + \frac{108 L (1-p)}{p^2} \zeta_{\star}^2 \eta^2.$$

Using Lemma 15 in Koloskova et al. (2020b) and $\eta \leq \frac{p}{60L}$, we can conclude the statement. $\qquad \square$

### C.5 PROOF IN NON-CONVEX CASE

**Lemma 2.** *Suppose that Assumptions 1 and 3 hold. It holds that*

$$\frac{1}{n}\mathbb{E}\left\|\left(\nabla F(\boldsymbol{X}^{(r)};\xi^{(r)}) - \nabla F(\boldsymbol{X}^{(r)})\right)\left(\boldsymbol{W} - \frac{1}{n}\mathbf{1}\mathbf{1}^\top\right)\right\|_F^2 \leq \frac{\sigma^2}{n}\sum_{i=2}^n \lambda_i^2. \tag{4}$$

*Note that $\frac{1}{n}\sum_{i=2}^n \lambda_i^2 \leq \max_{i\geq 2}(\lambda_i^2) < 1$.*

*Proof.* The statement follows from Assumptions 1 and 3 and Lemma 6. $\square$

**Lemma 15.** *Suppose that Assumptions 1 to 4 hold. When $\eta \leq \frac{1}{4L}$, it holds that*

$$\mathbb{E}f(\bar{\boldsymbol{x}}^{(r+1)}) \leq \mathbb{E}f(\bar{\boldsymbol{x}}^{(r)}) - \frac{\eta}{4}\left\|\nabla f(\bar{\boldsymbol{x}}^{(r)})\right\|^2 + \eta L^2 \Xi^{(r,0)} + \frac{L\sigma^2\eta^2}{n}. \tag{13}$$

*Proof.* See Lemma 11 in Koloskova et al. (2020b). $\square$

**Lemma 3.** *Suppose that Assumptions 1 to 4 hold. Then, when $\eta \leq \frac{p}{5L}$, it holds that*

$$\Xi^{(r+1,k)} \leq \left(1+\frac{p}{2}\right)\Xi^{(r,k+1)} + \frac{p}{4}(1-p)^{k+1}\Xi^{(r,0)} + \frac{6\zeta^2\eta^2}{p}(1-p)^{k+1} + \frac{\sigma^2\eta^2}{n}\sum_{i=2}^n \lambda_i^{2(k+1)}, \tag{7}$$

*where $\Xi^{(r,k)} := \frac{1}{n}\mathbb{E}\left\|\boldsymbol{X}^{(r)}\boldsymbol{W}^k - \bar{\boldsymbol{X}}^{(r)}\right\|_F^2$.*

*Proof.* We have

$$\mathbb{E}_r\left\|\boldsymbol{X}^{(r+1)}\boldsymbol{W}^k - \bar{\boldsymbol{X}}^{(r+1)}\right\|_F^2$$

$$= \mathbb{E}_r\left\|\boldsymbol{X}^{(r)}\boldsymbol{W}^{k+1} - \bar{\boldsymbol{X}}^{(r)} - \eta\nabla F(\boldsymbol{X}^{(r)};\xi^{(r)})\boldsymbol{W}^{k+1} + \eta\nabla F(\boldsymbol{X}^{(r)};\xi^{(r)})\frac{1}{n}\mathbf{1}\mathbf{1}^\top\right\|_F^2$$

$$= \underbrace{\left\|\boldsymbol{X}^{(r)}\boldsymbol{W}^{k+1} - \bar{\boldsymbol{X}}^{(r)} - \eta\nabla F(\boldsymbol{X}^{(r)})\boldsymbol{W}^{k+1} + \eta\nabla F(\boldsymbol{X}^{(r)})\frac{1}{n}\mathbf{1}\mathbf{1}^\top\right\|_F^2}_{T_1}$$

$$+ \eta^2 \underbrace{\mathbb{E}_r\left\|\left(\nabla F(\boldsymbol{X}^{(r)};\xi^{(r)}) - \nabla F(\boldsymbol{X}^{(r)})\right)\left(\boldsymbol{W}^{k+1} - \frac{1}{n}\mathbf{1}\mathbf{1}^\top\right)\right\|_F^2}_{T_2}.$$

$T_1$ and $T_2$ are bounded as follows.

$$T_1 \leq (1+\frac{p}{2})\left\|\boldsymbol{X}^{(r)}\boldsymbol{W}^{k+1} - \bar{\boldsymbol{X}}^{(r)}\right\|_F^2 + (1+\frac{2}{p})\eta^2\left\|\nabla F(\boldsymbol{X}^{(r)})\left(\boldsymbol{W}^{k+1} - \frac{1}{n}\mathbf{1}\mathbf{1}^\top\right)\right\|_F^2$$

$$\leq (1+\frac{p}{2})\left\|\boldsymbol{X}^{(r)}\boldsymbol{W}^{k+1} - \bar{\boldsymbol{X}}^{(r)}\right\|_F^2 + \frac{3}{p}\eta^2\left\|\left(\nabla F(\boldsymbol{X}^{(r)}) \pm \nabla F(\bar{\boldsymbol{X}}^{(r)}) - \nabla f(\bar{\boldsymbol{X}}^{(r)})\right)\left(\boldsymbol{W}^{k+1} - \frac{1}{n}\mathbf{1}\mathbf{1}^\top\right)\right\|_F^2$$

$$\leq (1+\frac{p}{2})\left\|\boldsymbol{X}^{(r)}\boldsymbol{W}^{k+1} - \bar{\boldsymbol{X}}^{(r)}\right\|_F^2 + \frac{3}{p}\eta^2\left\|\boldsymbol{W}^{k+1} - \frac{1}{n}\mathbf{1}\mathbf{1}^\top\right\|_{\text{op}}^2\left\|\nabla F(\boldsymbol{X}^{(r)}) \pm \nabla F(\bar{\boldsymbol{X}}^{(r)}) - \nabla f(\bar{\boldsymbol{X}}^{(r)})\right\|_F^2$$

$$\leq (1+\frac{p}{2})\left\|\boldsymbol{X}^{(r)}\boldsymbol{W}^{k+1} - \bar{\boldsymbol{X}}^{(r)}\right\|_F^2 + \frac{3}{p}\eta^2(1-p)^{k+1}\left\|\nabla F(\boldsymbol{X}^{(r)}) \pm \nabla F(\bar{\boldsymbol{X}}^{(r)}) - \nabla f(\bar{\boldsymbol{X}}^{(r)})\right\|_F^2$$

$$\leq (1+\frac{p}{2})\left\|\boldsymbol{X}^{(r)}\boldsymbol{W}^{k+1} - \bar{\boldsymbol{X}}^{(r)}\right\|_F^2 + \frac{6}{p}\eta^2(1-p)^{k+1}\left\|\nabla F(\boldsymbol{X}^{(r)}) - \nabla F(\bar{\boldsymbol{X}}^{(r)})\right\|_F^2$$

$$+ \frac{6}{p}\eta^2(1-p)^{k+1}\left\|\nabla F(\bar{\boldsymbol{X}}^{(r)}) - \nabla f(\bar{\boldsymbol{X}}^{(r)})\right\|_F^2$$

$$\leq (1+\frac{p}{2})\left\|\boldsymbol{X}^{(r)}\boldsymbol{W}^{k+1} - \bar{\boldsymbol{X}}^{(r)}\right\|_F^2 + \frac{6L^2}{p}\eta^2(1-p)^{k+1}\left\|\boldsymbol{X}^{(r)} - \bar{\boldsymbol{X}}^{(r)}\right\|_F^2 + \frac{6n\zeta^2}{p}\eta^2(1-p)^{k+1},$$

where we use $\|\boldsymbol{W}^{k+1} - \frac{1}{n}\mathbf{1}\mathbf{1}^\top\|_{\mathrm{op}}^2 = (1-p)^{k+1}$ in the last inequality. Using Lemma 6 and Assumption 3, we have

$$T_2 \leq \sigma^2 \sum_{i=1}^{n} \lambda_i^{2(k+1)}.$$

By combining the upper bounds of $T_1$ and $T_2$, we obtain

$$\Xi^{(r+1,k)} \leq (1+\frac{p}{2})\Xi^{(r,k+1)} + \frac{6L^2}{p}\eta^2(1-p)^{k+1}\Xi^{(r,0)} + \frac{6\zeta^2}{p}\eta^2(1-p)^{k+1} + \frac{\sigma^2\eta^2}{n}\sum_{i=1}^{n}\lambda_i^{2(k+1)}.$$

Using $\eta \leq \frac{p}{5L}$, we can conclude the statement. $\qquad\square$

**Lemma 16.** *Suppose that Assumptions 1 to 4 hold. When $\eta \leq \frac{p}{5L}$, it holds that*

$$\Xi^{(r,k)} \leq C(r,k) + \frac{6\zeta^2\eta^2}{p}(1-p)^{k+1}\sum_{r'=0}^{r-1}\left((1-p)(1+\frac{3p}{4})\right)^{r-r'-1}$$

$$+ \frac{p}{4}(1-p)^{k+1}\sum_{r'=1}^{r-1}\left((1-p)(1+\frac{3p}{4})\right)^{r-r'-1}C(r',0), \qquad (14)$$

*where*

$$C(r,k) := \frac{\sigma^2\eta^2}{n}\sum_{i=2}^{n}\lambda_i^{2(k+1)}\sum_{r'=0}^{r-1}\left(\lambda_i^2(1+\frac{p}{2})\right)^{r'}.$$

*Proof.* When $r = 1$, Eq. (14) holds from Lemma 3 and $\Xi^{(0,k)} = 0$ for all $k$.

Suppose that Eq. (14) holds when $r = r''$. We have

$$\Xi^{(r''+1,k)} \leq (1+\frac{p}{2})\Xi^{(r'',k+1)} + \frac{p}{4}(1-p)^{k+1}\Xi^{(r'',0)} + \frac{6\zeta^2}{p}\eta^2(1-p)^{k+1} + \frac{\sigma^2\eta^2}{n}\sum_{i=1}^{n}\lambda_i^{2(k+1)}$$

$$\leq (1+\frac{p}{2})C(r'',k+1) + \frac{\sigma^2\eta^2}{n}\sum_{i=1}^{n}\lambda_i^{2(k+1)} + \frac{6\zeta^2\eta^2}{p}(1-p)^{k+1}\sum_{r'=0}^{r''}\left((1-p)(1+\frac{3p}{4})\right)^{r''-r'}$$

$$+ \frac{p}{4}(1-p)^{k+1}\sum_{r'=1}^{r''}\left((1-p)(1+\frac{3p}{4})\right)^{r''-r'}C(r',0),$$

where we use Lemma 3 in the first inequality and the assumption that Eq. (14) holds when $r = r''$ in the second inequality. Using

$$\left(1+\frac{p}{2}\right)C(r'',k+1) + \frac{\sigma^2\eta^2}{n}\sum_{i=2}^{n}\lambda_i^{2(k+1)} = C(r''+1,k),$$

we have

$$\Xi^{(r''+1,k)} \leq C(r''+1,k) + \frac{6\zeta^2\eta^2}{p}(1-p)^{k+1}\sum_{r'=0}^{r''}\left((1-p)(1+\frac{3p}{4})\right)^{r''-r'}$$

$$+ \frac{p}{4}(1-p)^{k+1}\sum_{r'=1}^{r''}\left((1-p)(1+\frac{3p}{4})\right)^{r''-r'}C(r',0)$$

Thus, Eq. (14) holds when $r = r'' + 1$. From mathematical induction, we can conclude the statement. $\qquad\square$

**Lemma 4.** *Suppose that Assumptions 1 to 4 hold, and $\{\boldsymbol{x}_i^{(0)}\}_{i=1}^{n}$ are initialized to the same value. When $\eta \leq \frac{p}{5L}$, it holds that*

$$\Xi^{(r)} = \Xi^{(r,0)} \leq \frac{24\zeta^2(1-p)}{p^2}\eta^2 + 3\sigma^2\left(\frac{1}{n}\sum_{i=2}^{n}\frac{\lambda_i^2}{1-\lambda_i^2}\right)\eta^2. \qquad (8)$$

*Proof.* From Lemma 16, we have

$$\Xi^{(r,0)} \le C(r,0) + \underbrace{\frac{6\zeta^2\eta^2}{p}(1-p)\sum_{r'=0}^{r-1}\left((1-p)(1+\frac{3p}{4})\right)^{r-r'-1}}_{T_1}$$

$$+ \underbrace{\frac{p}{4}(1-p)\sum_{r'=1}^{r-1}\left((1-p)(1+\frac{3p}{4})\right)^{r-r'-1}C(r',0)}_{T_2},$$

$$C(r,0) = \frac{\sigma^2\eta^2}{n}\sum_{i=2}^{n}\lambda_i^2\sum_{r'=0}^{r-1}\left(\lambda_i^2(1+\frac{p}{2})\right)^{r'}$$

$$\le \frac{\sigma^2\eta^2}{n}\sum_{i=2}^{n}\lambda_i^2\sum_{r'=0}^{r-1}\left(1-\frac{1-\lambda_i^2}{2}\right)^{r'}$$

$$\le \frac{2\sigma^2\eta^2}{n}\sum_{i=2}^{n}\frac{\lambda_i^2}{1-\lambda_i^2},$$

where we use $p = 1 - \max_{i\ge 2}(\lambda_i^2) \le 1 - \lambda_i^2$ in the first inequality.

$$T_1 \le \frac{6\zeta^2\eta^2}{p}(1-p)\sum_{r'=0}^{r-1}\left(1-\frac{p}{4}\right)^{r-r'-1} \le \frac{24\zeta^2\eta^2}{p^2}(1-p).$$

$$T_2 = \frac{p\sigma^2\eta^2}{4n}(1-p)\sum_{i=2}^{n}\lambda_i^2\sum_{r'=1}^{r-1}\sum_{r''=0}^{r'-1}\left((1-p)(1+\frac{3p}{4})\right)^{r-r'-1}\left(\lambda_i^2(1+\frac{p}{2})\right)^{r''}$$

$$= \frac{p\sigma^2\eta^2}{4n}(1-p)\sum_{i=2}^{n}\lambda_i^2\sum_{r''=0}^{r-2}\left(\lambda_i^2(1+\frac{p}{2})\right)^{r''}\sum_{r'=r''+1}^{r-1}\left((1-p)(1+\frac{3p}{4})\right)^{r-r'-1}$$

$$\le \frac{p\sigma^2\eta^2}{4n}(1-p)\sum_{i=2}^{n}\lambda_i^2\sum_{r''=0}^{r-2}\left(\lambda_i^2(1+\frac{p}{2})\right)^{r''}\sum_{r'=r''+1}^{r-1}\left(1-\frac{p}{4}\right)^{r-r'-1}$$

$$\le \frac{\sigma^2\eta^2}{n}(1-p)\sum_{i=2}^{n}\lambda_i^2\sum_{r''=0}^{r-2}\left(\lambda_i^2(1+\frac{p}{2})\right)^{r''}$$

$$\le \frac{\sigma^2\eta^2}{n}(1-p)\sum_{i=2}^{n}\lambda_i^2\sum_{r''=0}^{r-2}\left(\lambda_i^2(1+\frac{p}{2})\right)^{r''}$$

$$\le \frac{\sigma^2\eta^2}{n}(1-p)\sum_{i=2}^{n}\lambda_i^2\sum_{r''=0}^{r-2}\left(1-\frac{1-\lambda_i^2}{2}\right)^{r''}$$

$$\le \frac{\sigma^2\eta^2}{n}(1-p)\sum_{i=2}^{n}\frac{\lambda_i^2}{1-\lambda_i^2}.$$

Combining the above three inequalities, we can obtain the desired results. □

**Lemma 17.** *Suppose that Assumptions 1 to 4 hold, and $\{\boldsymbol{x}_i^{(0)}\}_{i=1}^n$ are initialized to the same value. Then, there exists $\eta \le \frac{p}{5L}$ such that*

$$\frac{1}{R+1}\sum_{r=0}^{R}\mathbb{E}\left\|\nabla f(\bar{\boldsymbol{x}}^{(r)})\right\|^2$$

$$\le \mathcal{O}\left(\sqrt{\frac{L\sigma^2 F_0}{nR}} + \left(\left(\frac{\sigma^2}{n}\sum_{i=2}^{n}\frac{\lambda_i^2}{1-\lambda_i^2} + \frac{(1-p)\zeta^2}{p^2}\right)\frac{L^2 F_0^2}{R^2}\right)^{\frac{1}{3}} + \frac{LF_0}{Rp}\right),$$

*where $F_0 := f(\bar{\boldsymbol{x}}^{(0)}) - f^\star$.*

*Proof.* From Lemma 15, we have

$$\frac{1}{4(R+1)} \sum_{r=0}^{R} \mathbb{E} \left\| \nabla f(\bar{\boldsymbol{x}}^{(r)}) \right\|^2 \leq \frac{f(\bar{\boldsymbol{x}}^{(0)}) - f^\star}{\eta(R+1)} + \frac{L^2}{R+1} \sum_{r=0}^{R} \Xi^{(r,0)} + \frac{L\sigma^2\eta}{n}.$$

Using Lemma 4, we obtain

$$\frac{1}{4(R+1)} \sum_{r=0}^{R} \mathbb{E} \left\| \nabla f(\bar{\boldsymbol{x}}^{(r)}) \right\|^2 \leq \frac{f(\bar{\boldsymbol{x}}^{(0)}) - f^\star}{\eta(R+1)} + L^2 \left( \frac{3\sigma^2}{n} \sum_{i=2}^{n} \frac{\lambda_i^2}{1 - \lambda_i^2} + \frac{24(1-p)\zeta^2}{p^2} \right) \eta^2 + \frac{L\sigma^2}{n}\eta.$$

Using Lemma 17 in Koloskova et al. (2020b) and $\eta \leq \frac{p}{5L}$, we can conclude the statement.

$\square$

# D    PROOF OF THEOREM 2

In this section, we use the same notation introduced in Section C.2.

**Lemma 18.** *Suppose that Assumptions 1, 2 and 6 hold, $f_i = f_j$, and $\{\boldsymbol{x}_i^{(0)}\}_{i=1}^n$ are initialized to the same value. There exists $\eta \leq \frac{1}{24L}$ such that*

$$\frac{1}{R+1} \sum_{r=0}^{R} \left( \mathbb{E}f(\bar{\boldsymbol{x}}^{(r)}) - f^\star \right) \leq \mathcal{O}\left( \sqrt{\frac{\sigma_\star^2 r_0}{R}} + \frac{Lr_0}{R} \right),$$

*where $r_0 := \left\| \boldsymbol{x}^{(0)} - \boldsymbol{x}^\star \right\|^2$.*

*Proof.* We have

$$
\begin{aligned}
\mathbb{E}_r \left\| \boldsymbol{X}^{(r+1)} - \boldsymbol{X}^\star \right\|_F^2 &= \mathbb{E}_r \left\| \boldsymbol{X}^{(r)}\boldsymbol{W} - \eta \nabla F(\boldsymbol{X}^{(r)}; \xi^{(r)})\boldsymbol{W} - \boldsymbol{X}^\star \right\|_F^2 \\
&\leq \|\boldsymbol{W}\|_{\text{op}}^2 \, \mathbb{E}_r \left\| \boldsymbol{X}^{(r)} - \eta \nabla F(\boldsymbol{X}^{(r)}; \xi^{(r)}) - \boldsymbol{X}^\star \right\|_F^2 \\
&= \mathbb{E}_r \left\| \boldsymbol{X}^{(r)} - \eta \nabla F(\boldsymbol{X}^{(r)}; \xi^{(r)}) - \boldsymbol{X}^\star \right\|_F^2 \\
&= \underbrace{\left\| \boldsymbol{X}^{(r)} - \eta \nabla f(\boldsymbol{X}^{(r)}) - \boldsymbol{X}^\star \right\|_F^2}_{T_1} + \eta^2 \underbrace{\mathbb{E}_r \left\| \nabla F(\boldsymbol{X}^{(r)}; \xi^{(r)}) - \nabla f(\boldsymbol{X}^{(r)}) \right\|_F^2}_{T_2},
\end{aligned}
$$

where we use $\|\boldsymbol{W}\|_{\text{op}} = 1$ in the second equality. $T_1$ is bounded from above as follows:

$$
\begin{aligned}
T_1 &= \left\| \boldsymbol{X}^{(r)} - \boldsymbol{X}^\star \right\|_F^2 - 2\eta \left\langle \boldsymbol{X}^{(r)} - \boldsymbol{X}^\star, \nabla f(\boldsymbol{X}^{(r)}) \right\rangle + \eta^2 \left\| \nabla f(\boldsymbol{X}^{(r)}) \right\|_F^2 \\
&\leq \left\| \boldsymbol{X}^{(r)} - \boldsymbol{X}^\star \right\|_F^2 + (2\eta - L\eta^2) \left\langle \boldsymbol{X}^\star - \boldsymbol{X}^{(r)}, \nabla f(\boldsymbol{X}^{(r)}) \right\rangle.
\end{aligned}
$$

Furthermore, using $\langle \boldsymbol{x}^\star - \boldsymbol{x}, \nabla f(\boldsymbol{x}) \rangle \leq 0$ and $\eta \leq \frac{1}{L}$, we have

$$T_1 \leq \left\| \boldsymbol{X}^{(r)} - \boldsymbol{X}^\star \right\|_F^2 - \eta \sum_{i=1}^n \left( f(\boldsymbol{x}_i^{(r)}) - f^\star \right).$$

$T_2$ is bounded from above as follows:

$$
\begin{aligned}
T_2 &\leq 3\mathbb{E}_r \left\| \nabla F(\boldsymbol{X}^{(r)}; \xi^{(r)}) - \nabla F(\boldsymbol{X}^\star; \xi^{(r)}) \right\|_F^2 + 3 \left\| \nabla f(\boldsymbol{X}^{(r)}) \right\|_F^2 + 3\mathbb{E}_r \left\| \nabla F(\boldsymbol{X}^\star; \xi^{(r)}) \right\|_F^2 \\
&\leq 12L \sum_{i=1}^n \left( f(\boldsymbol{x}_i^{(r)}) - f^\star \right) + 3n\sigma_\star^2.
\end{aligned}
$$

Combining the above two inequalities, we obtain

$$\mathbb{E}_r \left\| \boldsymbol{X}^{(r+1)} - \boldsymbol{X}^\star \right\|_F^2 \leq \left\| \boldsymbol{X}^{(r)} - \boldsymbol{X}^\star \right\|_F^2 + 3n\sigma_\star^2\eta^2 - (\eta - 12L\eta^2) \sum_{i=1}^n \left( f(\boldsymbol{x}_i^{(r)}) - f^\star \right).$$

Using $\eta \leq \frac{1}{24L}$, we can obtain

$$\mathbb{E}_r \left\| \boldsymbol{X}^{(r+1)} - \boldsymbol{X}^\star \right\|_F^2 \leq \left\| \boldsymbol{X}^{(r)} - \boldsymbol{X}^\star \right\|_F^2 + 3n\sigma_\star^2\eta^2 - \frac{\eta}{2} \sum_{i=1}^n \left( f(\boldsymbol{x}_i^{(r)}) - f^\star \right).$$

Thus, we obtain

$$\frac{2}{n} \sum_{i=1}^n \left( \mathbb{E}f(\boldsymbol{x}_i^{(r)}) - f^\star \right) \leq \frac{\mathbb{E} \left\| \boldsymbol{X}^{(r)} - \boldsymbol{X}^\star \right\|_F^2 - \mathbb{E} \left\| \boldsymbol{X}^{(r+1)} - \boldsymbol{X}^\star \right\|_F^2}{n\eta} + 3\sigma_\star^2\eta.$$

$$\frac{2}{n(R+1)} \sum_{r=0}^{R} \sum_{i=1}^{n} \left( \mathbb{E} f(\boldsymbol{x}_i^{(r)}) - f^\star \right) \leq \frac{\left\| \boldsymbol{x}^{(0)} - \boldsymbol{x}^\star \right\|^2}{\eta(R+1)} + 3\sigma_\star^2 \eta.$$

Tuning the stepsize as in Lemma 17 in (Koloskova et al., 2020b), we obtain

$$\frac{2}{n(R+1)} \sum_{r=0}^{R} \sum_{i=1}^{n} \left( \mathbb{E} f(\boldsymbol{x}_i^{(r)}) - f^\star \right) \leq \mathcal{O}\left( \sqrt{\frac{\sigma_\star^2 \left\| \boldsymbol{x}^{(0)} - \boldsymbol{x}^\star \right\|^2}{R}} + \frac{L \left\| \boldsymbol{x}^{(0)} - \boldsymbol{x}^\star \right\|^2}{R} \right)$$

Finally, using $f(\bar{\boldsymbol{x}}) \leq \frac{1}{n} \sum_{i=1}^{n} f(\boldsymbol{x}_i)$, we can conclude the statement. $\qquad\square$

**Lemma 19.** *Suppose that Assumptions 1, 2, 5 and 6 hold, $f_i = f_j$ for all $i$ and $j$, and $\{\boldsymbol{x}_i^{(0)}\}_{i=1}^{n}$ are initialized to the same value. Then, there exists $\eta$ such that $\frac{1}{R+1} \sum_{r=0}^{R} (\mathbb{E} f(\bar{\boldsymbol{x}}^{(r)}) - f(\boldsymbol{x}^\star))$ is bounded from above by*

$$\mathcal{O}\left( \min\left\{ \sqrt{\frac{r_0 \sigma_\star^2}{nR}} + \left( \left( \frac{1}{n} \sum_{i=2}^{n} \frac{\lambda_i^2}{1 - \lambda_i^2} \right) \frac{L r_0^2 \sigma_\star^2}{R^2} \right)^{\frac{1}{3}} + \frac{L r_0}{Rp}, \quad \sqrt{\frac{r_0 \sigma_\star^2}{R}} + \frac{L r_0}{R} \right\} \right),$$

*where $\bar{\boldsymbol{x}}^{(r)} := \frac{1}{n} \sum_{i=1}^{n} \boldsymbol{x}_i^{(r)}$ and $r_0 := \|\bar{\boldsymbol{x}}^{(0)} - \boldsymbol{x}^\star\|^2$.*

*Proof.* The statement follows from Theorem 1 and Lemma 18. $\qquad\square$

## E  DERIVATION OF TRANSIENT ITERATIONS

Table 2: Comparison between $\frac{1}{p}$ and $\frac{1}{n}\sum_{i=2}^{n}\frac{\lambda_i^2}{1-\lambda_i^2}$. We numerically estimate $\frac{1}{n}\sum_{i=2}^{n}\frac{\lambda_i^2}{1-\lambda_i^2}$ from Fig. 1. $\frac{1}{p}$ has been derived by Nedić et al. (2018).

|  | Ring | Torus | Hypercube |
|---|---|---|---|
| $\frac{1}{p}$ | $\mathcal{O}(n^2)$ | $\mathcal{O}(n)$ | $\mathcal{O}(\log(n))$ |
| $\frac{1}{n}\sum_{i=2}^{n}\frac{\lambda_i^2}{1-\lambda_i^2}$ | $\mathcal{O}(n)$ | $\mathcal{O}(\log(n))$ | $\mathcal{O}(1)$ |

**Transient Iterations of Theorem 1:**  From Theorem 1, when $\zeta = 0$, Decentralized SGD can achieve the following convergence rate:

$$\frac{1}{R+1}\sum_{r=0}^{R}\left\|\nabla f(\bar{\boldsymbol{x}}^{(r)})\right\|^2 \leq \mathcal{O}\left(\sqrt{\frac{L\sigma^2 F_0}{nR}} + \left(\left(\frac{\sigma^2}{n}\sum_{i=2}^{n}\frac{\lambda_i^2}{1-\lambda_i^2}\right)\frac{L^2 F_0^2}{R^2}\right)^{\frac{1}{3}} + \frac{LF_0}{Rp}\right).$$

Thus, to satisfy $\frac{1}{R+1}\sum_{r=0}^{R}\left\|\nabla f(\bar{\boldsymbol{x}}^{(r)})\right\|^2 \leq \mathcal{O}(\sqrt{\frac{L\sigma^2 F_0}{nR}})$, it requires

$$R \geq \max\left\{n^3\left(\frac{1}{n}\sum_{i=2}^{n}\frac{\lambda_i^2}{1-\lambda_i^2}\right)^2, \frac{n}{p^2}\right\}\frac{LF_0}{\sigma^2}.$$

Using Table 2, we can derive the results shown in Table 1.

**Transient Iterations of Proposition 1:**  From Proposition 1, when $\zeta = 0$, Decentralized SGD can achieve the following convergence rate:

$$\frac{1}{R+1}\sum_{r=0}^{R}\left\|\nabla f(\bar{\boldsymbol{x}}^{(r)})\right\|^2 \leq \mathcal{O}\left(\sqrt{\frac{L\sigma^2 F_0}{nR}} + \left(\left(\frac{(1-p)\sigma^2}{p}\right)\frac{L^2 F_0^2}{R^2}\right)^{\frac{1}{3}} + \frac{LF_0}{Rp}\right).$$

Thus, to satisfy $\frac{1}{R+1}\sum_{r=0}^{R}\left\|\nabla f(\bar{\boldsymbol{x}}^{(r)})\right\|^2 \leq \mathcal{O}(\sqrt{\frac{L\sigma^2 F_0}{nR}})$, it requires

$$R \geq \max\left\{\frac{(1-p)^2 n^3}{p^2}, \frac{n}{p^2}\right\}\frac{LF_0}{\sigma^2}.$$

Using Table 2, we can derive the results shown in Table 1.

## F  EXPERIMENTAL SETTING

In this section, we explain in detail how to create the graphs used in Section 6.2. In Section 6.2, we generated the topologies as follows:

1. Let $\boldsymbol{W}$ be the mixing matrix of a base graph, such as the ring and torus.

2. We factor $\boldsymbol{W}$ as $\boldsymbol{W} = \boldsymbol{V}\Lambda\boldsymbol{V}^{-1}$ where $\boldsymbol{V}$ is the $n \times n$ matrix whose $i$-th column is the eigenvector $\boldsymbol{v}_i$ of $\boldsymbol{W}$, and $\Lambda$ is the $n \times n$ diagonal matrix whose diagonal elements are the corresponding eigenvalues, $\Lambda_{ii} = \lambda_i$. Without the loss of generality, we assume that $|\lambda_2| \geq |\lambda_j|$ for all $j \geq 2$.

3. We define $\tilde{\Lambda}$ as follows:

$$\tilde{\Lambda} = \begin{pmatrix} \lambda_1 & & & & \\ & \lambda_2 & & & \\ & & \lambda & & \\ & & & \ddots & \\ & & & & \lambda \end{pmatrix},$$

where $\lambda \in [-|\lambda_2|, |\lambda_2|]$ is the hyperparameter.

4. We generate a graph that has the mixing matrix of $\boldsymbol{V}\tilde{\Lambda}\boldsymbol{V}^{-1}$.

$\boldsymbol{V}\tilde{\Lambda}\boldsymbol{V}^{-1}$ has the same spectral gap $p$ as $\boldsymbol{W}$, while we can change $\frac{1}{n}\sum_{i=2}^{n}\frac{\lambda_i^2}{1-\lambda_i^2}$ by varying $\lambda$. Note that the inverse matrix $\boldsymbol{V}^{-1}$ does not always exist, but we numerically confirmed that the inverse matrix exists for the line, ring, and torus with $n = 200$.

## G  ADDITION NUMERICAL RESULTS

In Section 6.2, we used MNIST as the training dataset. The following figure shows the results with CIFAR-10, and we can observe the same trend as in MNIST.

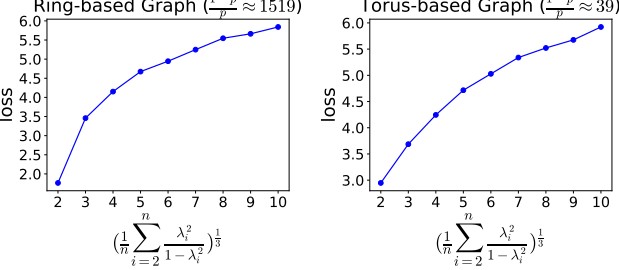

(a) Logistic Regression

Figure 3: The impact of $\frac{1}{n}\sum_{i=2}^{n}\left(\lambda_i^2/1-\lambda_i^2\right)$ on the loss value at the final parameter with CIFAR-10. In each figure, we vary the eigenvalues to construct different topologies while keeping $p$ constant. We observe that the loss value increases as $\frac{1}{n}\sum_{i=2}^{n}\left(\lambda_i^2/1-\lambda_i^2\right)$ increases, which is consistent with Theorems 1 and 2. Error bars are omitted since all standard errors were smaller than $1.0 \times 10^{-5}$ and visually indistinguishable.

## H  LLM USAGE

We used LLM for proofreading, and it did not contribute to the content of the paper itself.

