# OpenReview forum: "Revisiting the Effect of Topologies on Decentralized SGD"
_ICLR.cc/2026/Conference — Submitted to ICLR 2026_

### Official Review · Reviewer_CSwe · 2025-10-26

**Soundness:** 4
**Presentation:** 4
**Contribution:** 2
**Rating:** 6
**Confidence:** 4

**Summary:**

This paper proposed improved analysis on how network topologies affect convergence rates of decentralized SGD. Previous analysis mainly focus on spectral gap of communication networks, while this paper's analysis is based on all eigenvalues of gossip matrix, which leads to improved bounds including many classical topologies such as ring, torus and hypercube. Results for both convex and nonconvex objective functions are provided.

**Strengths:**

(1) The motivation is clear and strong. The new analysis based on eigenvalues of gossip matrix could be extended to more general settings.

(2) The proofs are technically sound and easy to read. The assumptions are standard.

**Weaknesses:**

(1) Only results for decentralized SGD are provided. It would be better to include results for more modern approaches such as gradient tracking.

(2) Analysis based on eigenvalues requires undirected graphs.

(3) It would be better to expand experiments on various tasks.

**Questions:**

(1) Is it possible to extend the analysis to directed graphs?

(2) Is it possible to derive any tightness results of the bounds in this paper?

---

> ### Author Response · Authors · 2025-11-26
>
> We thank the reviewer for your positive feedback.
>
> > Only results for decentralized SGD are provided. It would be better to include results for more modern approaches such as gradient tracking.
>
> The core technique for improving the convergence rate is the inequality presented in Lemma 2.
> This inequality is general and would be applicable to a wide range of decentralized learning methods; however, analyzing the other methods is beyond the scope of our paper.
>
> > Analysis based on eigenvalues requires undirected graphs.
>
> Thank you for your comments.
> When the mixing matrix $\mathbf{W}$ is doubly stochastic, it is easy to extend our analysis to directed graphs (i.e., $\mathbf{W}$ is not symmetric).
> In this case, its eigenvalues may be complex values, while we can rewrite equations with singular values of $\mathbf{W}$ instead of eigenvalues and can obtain similar results.
>
> When $\mathbf{W}$ is not doubly stochastic, it is not trivial to extend our results to directed graphs. However, we believe that the key lemma, Lemma 2, still plays an important role in improving the convergence rate.
>
> > It would be better to expand experiments on various tasks.
>
> Thank you for your comment.
> We added the results for CIFAR-10 in Sec. G, and the observation was consistent with the results of MNIST shown in Figure 2.
> Since the results for ridge regression are still in progress, we will add them in the camera-ready version once they are completed.
>
> > Is it possible to extend the analysis to directed graphs?
>
> See the above comments.
>
> > (2) Is it possible to derive any tightness results of the bounds in this paper?
>
> It is currently unknown whether our rate matches the lower bound.
> Although our newly derived theorem yields the best known convergence rate for Decentralized SGD to date, further improvements may still be possible.

---

### Official Review · Reviewer_nnfv · 2025-10-28

**Soundness:** 2
**Presentation:** 2
**Contribution:** 1
**Rating:** 2
**Confidence:** 4

**Summary:**

This paper presents a convergence analysis of Decentralized SGD, aiming to provide a more precise understanding of how communication topologies affect convergence. In my opinion, the contribution is incremental in both theory and experiment, and the work is not yet suitable for publication at ICLR.

**Strengths:**

This paper presents a convergence analysis of Decentralized SGD, aiming to provide a more precise understanding of how communication topologies affect convergence.

**Weaknesses:**

My main concerns are as follows:

1.	The paper lacks experimental results demonstrating the performance of Decentralized SGD under different topologies in both IID and non-IID settings, which are crucial to support the theoretical findings. Simply citing previous works or describing results in words is not convincing.

2.	In Proposition 1, the term $\frac{\xi^2}{p^2}$indicates that the impact of topology in the IID case is smaller than in the non-IID case, since $\xi = 0$ under IID data. The authors should more clearly explain their motivation to improve the analysis.

3.	The authors claim that, even in the non-IID case, the coefficient still depends on p. Can the authors provide a corresponding lower bound to justify this claim? Otherwise, one might question whether the analysis is tight and whether the coefficient should indeed depend on p in the non-IID setting.

4.	The presentation needs improvement. Instead of only giving proof sketches, the authors should provide more insight into why the coefficient can be improved in the IID case, what analytical challenges arise, and how they are addressed. At present, it is difficult to identify the main contributions, especially since the convergence of Decentralized SGD has been extensively studied.

5.	Is the analysis in the IID case tight? Can the coefficient be further improved?

6.	Is the proposed coefficient a proper measure for all topologies? The spectral gap is a widely used and well-understood measure that captures the influence of topology. The authors should justify whether their proposed coefficient is theoretically sound and more precise than the spectral gap. Additional theoretical and experimental results are needed to support this claim.

7.	Why does the hypercube topology in Figure 1 exhibit a different trend compared to the others?

8.	The experiments are insufficient. The authors should compare different topologies with the same number of nodes but varying communication links, and plot both the spectral gap and their proposed coefficient to more clearly illustrate the impact of topology.

**Questions:**

N/A

---

> ### Author Response · Authors · 2025-11-26
>
> Thank you for your comments.
> Based on your comments, we explained why improving the rate in the IID setting is important and provided intuition about the novel proof technique.
>
> We respectfully ask the reviewer to reconsider the evaluation and, if appropriate, adjust the score in light of these changes.
> If you have remaining concerns, we are happy to address them.
>
> > The paper lacks experimental results demonstrating the performance of Decentralized SGD under different topologies in both IID and non-IID settings [...] Simply citing previous works or describing results in words is not convincing.
>
> We appreciate the reviewer’s comment. However, extensive prior work has already evaluated Decentralized SGD under different topologies in both IID and non-IID settings [1,2,3,4], and they reported the consistent empirical trends that topologies have a little impact on the performance in the IID setting (see Figure 3 in [1], Figure 5b in [2], Figure 7a in [3], Figure 3a in [4], for example). Our contribution aims to explain these observed behaviors theoretically. Therefore we believe reproducing those results here is unnecessary.
>
> > In Proposition 1, the term $\frac{\zeta^2}{p^2}$ indicates that the impact of topology in the IID case is smaller than in the non-IID case, since $\zeta=0$ under IID data. The authors should more clearly explain their motivation to improve the analysis.
>
>
> While our theoretical improvements are significant in the IID case, we respectfully disagree that this makes the contribution marginal.
>
> Improving in the IID setting is important for the following reasons:
> * Many papers have observed that data heterogeneity does not significantly impact practical performance, e.g., [6], as real datasets are not typically too heterogeneous. Specifically, [6] reported that while FedAvg is theoretically exacerbated by high data heterogeneity, FedAvg converges pretty well on real heterogeneous datasets. We therefore would like to emphasize that improving the rate in the regime that $\zeta^2 \ll \sigma^2$ is still important.
> * Decentralized SGD is a core building block for many decentralized algorithms, and the proof techniques introduced in our analysis (e.g., Lemma 2) are not specific to Decentralized SGD. This approach may also be applicable to other decentralized learning methods.
> * Deriving a convergence rate that accurately captures the influence of the topology is valuable in its own right, as it can directly inform research on topology design and other practical applications.
>
>
>
> > The authors claim that, even in the non-IID case, the coefficient still depends on p. Can the authors provide a corresponding lower bound to justify this claim? [...]
>
>
> We agree that matching lower bounds are valuable. However, the fact that it is currently unknown whether our rate matches the lower bound does not diminish the significance of our contribution. As the reviewer also notes, the convergence rate of Decentralized SGD has been extensively studied, and our newly derived theorem yields the best known convergence rate for Decentralized SGD to date. While we acknowledge that further improvements may still be possible, our improvement represents a substantial step forward.
>
> > Instead of only giving proof sketches, the authors should provide more insight into why the coefficient can be improved in the IID case, what analytical challenges arise, and how they are addressed.
>
> We provided the following insight into why the coefficient of the stochastic noise $\sigma^2$ in our theorems in Sec. C.1.
>
> The most existing papers only used the spectral gap to measure the quality of topologies.
> The spectral gap $1 - \max_{i\geq2} (\lambda_i^2)$ satisfies the following inequality for any matrix $\mathbf{X}$ (see Lemma 1):
>
> $$
> \| \mathbf{X} \mathbf{W} - \bar{\mathbf{X}} \|^2_F \leq \max_{i\geq2} (\lambda_i^2) \| \mathbf{X} - \bar{\mathbf{X}} \|^2_F
> $$
>
> In this work, we found that if we restrict the input $X := (X_1, X_2, \dots, X_n)$ so that $X_1, \dots, X_n$ are independent random variables, have the same mean, and their variance is bounded by $\sigma^2$, we have (see Lemma 6):
>
> $$
> \mathbb{E} \| X \mathbf{W} - \bar{X} \|^2_F \leq \sigma^2 \sum_{i=2}^n \lambda_i^2
> $$
>
> The stochastic noise is independent among nodes. Thus, the coefficient of the stochastic noise $\sigma^2$ can be reduced to $\frac{1}{n} \sum_{i=2}^n \frac{\lambda_i^2}{1 - \lambda_i^2}$ from $\max_{i\geq2} (\lambda_i^2)$. This is the core idea for how the coefficient of $\sigma^2$ is improved in our theorems.

---

> ### Author Response · Authors · 2025-11-26
>
> > At present, it is difficult to identify the main contributions, especially since the convergence of Decentralized SGD has been extensively studied.
>
> **Our main contribution is improving the convergence rate of Decentralized SGD, achieving the best-known rate under standard assumptions.** As the reviewer pointed out, Decentralized SGD has been extensively studied. Thus, we believe it is significant that our work further improves the known convergence rates.
>
> > Is the analysis in the IID case tight? Can the coefficient be further improved?
>
> See the above response about the lower bound.
>
> > Is the proposed coefficient a proper measure for all topologies? [...]
>
> We make only Assumption 1 regarding topologies, namely that the mixing matrix $W$ is symmetric and doubly stochastic, as in prior works such as [5]. **We do not impose any additional assumptions regarding topologies.** In that sense, $\frac{1}{n} \sum_{i=2}^n \frac{\lambda_i^2}{1 - \lambda_i^2}$ is a proper quantity to measure the quality of topologies, and we show that $\frac{1}{n} \sum_{i=2}^n \frac{\lambda_i^2}{1 - \lambda_i^2} \leq \frac{1 - p}{p}$ in Eq. (3).
>
> Furthermore, proposing a measure beyond the spectral gap is a significant contribution. It not only improves analysis of Decentralized SGD but may also be useful in other fields where a topology plays a key role.
>
> > Why does the hypercube topology in Figure 1 exhibit a different trend compared to the others
>
> The hypercube is much denser than the other topologies, which allows information to mix quickly and keeps $\frac{1}{n} \sum_{i=1}^n \frac{\lambda_i^2}{1 - \lambda_i^2} = O(1)$, explaining the different trend.
>
> > The experiments are insufficient. The authors should compare different topologies with the same number of nodes but varying communication links [...]
>
> With our experimental design, we wanted to decouple the effect of $p$ and our new quantity in experiments.
> If both $p$ and our new quantity change, it will be hard to interpret from the figures to what extent each of the two quantities affects convergence.
>
>
> ## Reference
>
> [1] Lian et. al., Can decentralized algorithms outperform centralized algorithms? a case study for decentralized parallel stochastic gradient descent, In NeurIPS 2017
>
> [2] Neglia et. al., Decentralized gradient methods: Does topology matter?, In AISTATS 2020
>
> [3] Takezawa et. al., Beyond exponential graph: Communication-efficient topologies for decentralized learning via finite-time convergence, In NeurIPS 2023
>
> [4] Takezawa et. al., Scalable decentralized learning with teleportation, In ICLR 2025
>
> [5] Koloskova et. al., A unified theory of decentralized SGD with changing topology and local updates, In ICML 2020
>
> [6] Wang et al., On the Unreasonable Effectiveness of Federated Averaging with Heterogeneous Data, In TMLR 2024

---

### Official Review · Reviewer_2mdZ · 2025-10-28

**Soundness:** 2
**Presentation:** 3
**Contribution:** 1
**Rating:** 2
**Confidence:** 4

**Summary:**

This paper provides a tighter convergence analysis of the optimization error in the Decentralized SGD algorithm. The analysis shows that not only the spectral gap of the gossip matrix matters, but the entire spectrum also plays a role. While this insight does not directly improve the worst-case upper bounds in general heterogeneous settings, it does lead to better rates in homogeneous scenarios. This is clearly demonstrated in Table 1, where the authors compare the transient time (i.e., the number of iterations required to achieve linear speedup) of the state-of-the-art result by Koloskova et al. with their own findings across different graph topologies. The authors further argue that their results help explain why, in homogeneous settings, the choice of graph appears to have less impact. Finally, they support their theoretical findings with a set of experiments.

**Strengths:**

The paper is well written and enjoyable to read. The question of how the communication graph impacts decentralized learning, although extensively explored, remains highly relevant, and demonstrating that the entire spectrum of $W$ matters is a natural yet insightful contribution, which could be useful for the future design of communication topologies. I also appreciated the results presented in Table 1, which clearly illustrate the improvement of the proposed upper bounds over those of Koloskova et al.

**Weaknesses:**

As currently written, the paper feels somewhat premature for ICLR. The question of the communication graph’s impact is interesting and relevant, but the authors’ improvement over Koloskova et al. mainly applies to homogeneous settings, which are less central in decentralized learning than heterogeneous ones. The manuscript would be substantially strengthened by addressing the following points:

1) **Strongly convex case.** Add results for strongly convex objectives so the paper provides a complete improvement over Koloskova et al. across common problem classes.

2) **More intuition on the spectrum.** Explain more precisely what the full spectrum of $W$ captures and why it primarily affects the stochastic-variance term of the decentralization error (the middle term). Intuition, toy examples, or a short illustrative derivation would help readers understand the phenomenon.

3) **Richer experiments.** As noted, Fig. 2 omits error bars because they were “too small”: this suggests that the experiments may be too simple (MNIST). I recommend running experiments on more challenging benchmarks (e.g., CIFAR-10 and other realistic datasets) and adding cases that show a graph with a worse spectral gap can outperform one with a better gap thanks to a more favorable spectrum.

4) **Formalize Table 2.** Provide a formal derivation or rigorous bounds for $\frac{1}{n}\sum_i \frac{\lambda_i^2}{1-\lambda_i^2}$ reported in Table 2; currently these quantities are only estimated.

5) **Graph learning discussion.** Include or discuss in more depth a method for learning the communication graph (currently deferred to future work). Given the paper’s claims about spectrum-driven design, even a preliminary approach would increase the paper’s impact.

**Questions:**

Other comments/questions.

1) What exactly changes in the graph structure when $\frac{1}{n}\sum_i \frac{\lambda_i^2}{1-\lambda_i^2}$ varies, while keeping the spectral gap fixed (Fig.2)? Clarifying this point would help readers build intuition about how this spectral quantity reflects graph topology (see also point 2 in the main review).


2) The conclusion is sometimes misleading. The authors claim that their results explain why the graph does not have a large impact in homogeneous setting. But in fact it does (the obtained bounds are still vacuous for $p$ close to 0), it is simply provably less impactful.

3) In convex settings, $x^*$ may not be unique. How do you handle that in Assumptions 6-7 and in your proof?

4) In line 459, you mention “the same training dataset.” Could you clarify whether the datasets are truly identical across nodes, or if they share the same distribution but different realizations?

Typos:

- line 020: "[...] rate than the priori analysis" <-- "[...] rate **less** than the prior analysis"
- line 462: "See Section 6.2" <-- "See Appendix E"?

---

> ### Author Response · Authors · 2025-11-26
>
> Thank you for your constructive comments. Based on your comments, we added the convergence rate for the strongly convex case, provided intuition about the novel proof technique, and included the results for CIFAR-10. We agree that these additional results significantly strengthen our paper. We kindly ask the reviewer to reconsider the evaluation and, if possible, adjust the score to reflect these changes. If you have remaining concerns, we are happy to address them.
>
> > Add results for strongly convex objectives so the paper provides a complete improvement over Koloskova et al. across common problem classes.
>
> Thank you for the comments.
> **We analyzed the rate for the strongly convex case and added this result in Lemma 14 in the current version.**
> Similar to the convex and non-convex cases, we show that, in the strongly convex case, the coefficient of the stochastic gradient noise improves from $\frac{1}{p}$ to $\frac{1}{n} \sum_{i=2}^n \frac{\lambda_i^2}{1 - \lambda_i^2}$.
> We promise to add it to Theorem 1 in the camera-ready version.
>
>
> > Explain more precisely what the full spectrum of $W$ captures and why it primarily affects the stochastic-variance term of the decentralization error [...]
>
> Thank you for the comments.
> **We provided the following intuitive difference between the spectral gap and full spectral in Sec. C.1.**
>
> The spectral gap $1 - \max_{i\geq2} (\lambda_i^2)$ satisfies the following inequality **for any matrix** $\mathbf{X}$ (see Lemma 1):
>
> $$
> \| \mathbf{X} \mathbf{W} - \bar{\mathbf{X}} \|^2_F \leq \max_{i\geq2} (\lambda_i^2) \| \mathbf{X} - \bar{\mathbf{X}} \|^2_F
> $$
>
> On the other hand, if we restrict the input $X := (X_1, X_2, \dots, X_n)$ so that $X_1, \dots, X_n$ are **independent** random variables, have the same mean, and their variance is bounded by $\sigma^2$, we have (see Lemma 6):
>
> $$
> \mathbb{E} \| X \mathbf{W} - \bar{X} \|^2_F \leq \sigma^2 \sum_{i=2}^n \lambda_i^2
> $$
>
> **The stochastic noise is independent among nodes.** Thus, the coefficient of the stochastic noise $\sigma^2$ can be reduced to $\frac{1}{n} \sum_{i=2}^n \frac{\lambda_i^2}{1 - \lambda_i^2}$ from $\max_{i\geq2} (\lambda_i^2)$.
> This is an intuition of why the coefficient of $\sigma^2$ is improved in our theorem.
>
>
> > [...] I recommend running experiments on more challenging benchmarks (e.g., CIFAR-10 and other realistic datasets) [...]
>
> **We added the results with CIFAR-10 in Sec. G.**
> The observation was consistent with the results of MNIST.
> Since the results for ridge regression are still in progress, we will add them in the camera-ready version once they are completed.
>
>
> > Provide a formal derivation or rigorous bounds for $\frac{1}{n} \sum_i \frac{\lambda_i^2}{1 - \lambda_i^2}$ reported in Table 2; currently these quantities are only estimated.
>
> We agree with the reviewer that providing a formal proof would significantly strengthen our paper, though we believe that the current results are sufficient to support our main conclusions.
>
>
> > Include or discuss in more depth a method for learning the communication graph (currently deferred to future work). [...]
>
> Thank you for the insightful comment. **We included the following discussion in Sec. B.**
>
> We propose to generalize the approach of [1] to maximise our new quantity $\frac{1}{n} \sum_{i=2}^n \frac{\lambda_i^2}{1 - \lambda_i^2}$, instead of maximizing the spectral gap.
> More precisely, [1] proposed to decompose the original topology into matching in such a way that the spectral gap is maximized.  That allows at the same time to improve the spectral gap of the topology, and use less communications due to alternating between different matchings, providing improvement not only in the convergence speed, but also improving the runtime per iteration.
> We therefore propose to replace the maximization metric from the spectral gap to our new quantity $\frac{1}{n} \sum_{i=2}^n \frac{\lambda_i^2}{1 - \lambda_i^2}$, capturing tighter the practical behavior of the decentralized learning algorithms.
>
>
> [1] Wang et. al., MATCHA: Speeding up decentralized SGD via matching decomposition sampling. In arXiv, 2019.
>
> > The conclusion is sometimes misleading. [...]
>
> Thanks for pointing this out. We have revised the conclusion to make our statement more precise.
> If there are any remaining concerns or comments, we would be more than happy to make further improvements.
>
> > In convex settings, $x^\star$ may not be unique. How do you handle that in Assumptions 6-7 and in your proof?
>
> The statements hold for any optimal solution $x^\star$. We revised the statement of assumptions to clarify it.
>
> > In line 459, you mention “the same training dataset.” Could you clarify whether the datasets are truly identical across nodes, or if they share the same distribution but different realizations?
>
> In our experiments, all nodes share the same training datasets. We revised the text in Sec. 6.2 and clarified it in the revised manuscript.

---

### Official Review · Reviewer_fzn7 · 2025-11-02

**Soundness:** 2
**Presentation:** 3
**Contribution:** 2
**Rating:** 4
**Confidence:** 5

**Summary:**

This paper presents a refined convergence analysis of Decentralized Stochastic Gradient Descent (DSGD) under a given gossip communication scheme. While previous work primarily characterizes the convergence rate in terms of the spectral gap (i.e., the second-largest eigenvalue of the gossip matrix), this paper goes further by analyzing the influence of the entire spectrum of eigenvalues. The resulting convergence bound is tighter and reveals that, in the case of data-homogeneous decentralized learning, all eigenvalues—not just the second largest—significantly impact the convergence behavior. Theoretical insights are supported by experimental results that validate the findings.

**Strengths:**

- **Theoretical Contribution**: This paper provides a detailed theoretical analysis of the convergence behavior of Decentralized SGD by examining the full spectrum of eigenvalues $\lambda_2, \ldots, \lambda_n$ of the gossip matrix. The refined convergence bound, presented in Theorem 1, captures the influence of all eigenvalues, offering a more precise characterization compared to earlier results, which primarily rely on the spectral gap. For comparison, the classic convergence result is summarized in Proposition 1. The mathematical analysis is rigorous and well-supported.

- **Empirical Validation**: The numerical experiments compare the key convergence terms from previous work and this paper. The results clearly verify that the newly derived term is tighter, confirming the theoretical advantage.

**Weaknesses:**

This paper is mainly a theoretical paper. The major theorems are Theorem 1 and 2. However, there are limitations in both theorems.

- **Marginal Theoretical Improvement in Theorem 1**: In the main Theorem 1, Convex Case, the new rate improves the term
$$ \left(\frac{1-p }{p}\cdot\frac{\sigma^2Lr_0^2}{R^2}+\frac{(1-p)\zeta^2}{p^2}\cdot\frac{Lr_0^2}{R^2}\right)^{\frac{1}{3}}$$ to
$$\left(\frac{1}{n}\sum_{i=2}^n\frac{\lambda_i^2}{1-\lambda_i^2}\cdot\frac{\sigma^2Lr_0^2}{R^2} + \frac{(1-p)\zeta^2}{p^2}\cdot\frac{Lr_0^2}{R^2}\right)^{\frac{1}{3}}.$$
However, the new rate is only significantly tighter when $\sigma^2 \gg \frac{\zeta^2}{p}$, meaning that many of the theoretical advantages heavily rely on the data homogeneity assumption. The similar problem also lies in the Non-convex Case part of the theorem.

- **Overly Strong Assumptions in Theorem 2**: The assumptions in Theorem 2 are quite restrictive, requiring convexity and that all local functions are identical $(f_i = f_j$ for all $i, j$). Under such conditions, even running local SGD without communication would eventually lead each agent $x_i$ to converge to the same minimizer. Moreover, the convergence bound in Theorem 2 is given as the minimum of two terms:
The first term is essentially derived from Theorem 1 when $\zeta = 0$.
The second term reflects the performance of independent local SGD, which is intuitive and not surprising.
As a result, the contribution of Theorem 2 is somewhat limited.

**Questions:**

As I mentioned in the weakness on Theorem 1, Convex Case, the new rate improves the term $$ \left(\frac{1-p }{p}\cdot\frac{\sigma^2Lr_0^2}{R^2}+\frac{(1-p)\zeta^2}{p^2}\cdot\frac{Lr_0^2}{R^2}\right)^{\frac{1}{3}}$$ to $$\left(\frac{1}{n}\sum_{i=2}^n\frac{\lambda_i^2}{1-\lambda_i^2}\cdot\frac{\sigma^2Lr_0^2}{R^2} + \frac{(1-p)\zeta^2}{p^2}\cdot\frac{Lr_0^2}{R^2}\right)^{\frac{1}{3}}.$$

You've already presented the comparison between
$
\frac{1}{n}\sum_{i=2}^n \frac{\lambda_i^2}{1 - \lambda_i^2}
$
and $
\frac{1 - p}{p}.
$
Could you also include a comparison with
$
\frac{1 - p}{p^2}
$
in front of the $\zeta^2$ term? I would like to gain a clearer understanding of how the ratio $\frac{\sigma^2}{\zeta^2}$ affects the bound in Theorem 1 and under what conditions this bound becomes significantly tighter than the previous bound.

---

> ### Author Response · Authors · 2025-11-26
>
> Thank you for the comments.
> We addressed your concern that our contribution may be marginal because the improvement is most visible in the IID case. We clarified why the IID regime is also practically important and why the contribution is not marginal.
>
> We would appreciate it if this clarification could be reflected in the final score.
> If you have any remaining concerns or comments, we are happy to address them.
>
>
> > Marginal Theoretical Improvement in Theorem 1: ...
>
> While our theoretical improvements are indeed most significant in the regime where $\sigma^2 \gg \frac{\zeta^2}{p}$, we respectfully disagree that this makes the contribution marginal.
>
> Improving in the IID setting is important for the following reasons:
>
> * Many papers have observed that data heterogeneity does not significantly impact practical performance, e.g., [1], as real datasets are not typically too heterogeneous. Specifically, [1] reported that while FedAvg is theoretically exacerbated by high data heterogeneity, FedAvg converges pretty well on real heterogeneous datasets. We therefore would like to emphasize that improving the rate and understanding the convergence behavior in the regime that $\frac{\zeta^2}{p} \ll \sigma^2$ is still important.
> * Decentralized SGD is a core building block for many decentralized algorithms, and the proof techniques introduced in our analysis (e.g., Lemma 2) are not specific to Decentralized SGD. This approach may also be applicable to many other decentralized learning methods.
> * Deriving a convergence rate that accurately captures the influence of the topology is valuable in its own right, as it can directly inform research on practical applications, such as topology design. We also added a discussion of the preliminary approach to designing topologies using our new quantity $\frac{1}{n} \sum_{i=2}^n \frac{\lambda_i^2}{1 - \lambda_i^2}$ in Section B.
>
>
> > Overly Strong Assumptions in Theorem 2: ...
>
> The convergence rate under the general setting is provided in Theorem 1, and Theorem 2 is intended to show that, under stronger assumptions (i.e., the convex and homogeneous settings), one can obtain an improved convergence rate. Thus, we believe that Theorem 2 should not be viewed negatively, but instead as a complementary contribution.
>
>
> > Could you also include a comparison with $\frac{1 - p}{p}$ in front of the $\zeta^2$  [...]
>
> We are running the experiments. We promise to add these results in the camera-ready version.
>
> ## Reference
>
> [1] Wang et al., On the Unreasonable Effectiveness of Federated Averaging with Heterogeneous Data, In TMLR 2024

---

### Official Review · Reviewer_suPG · 2025-11-03

**Soundness:** 4
**Presentation:** 4
**Contribution:** 3
**Rating:** 6
**Confidence:** 3

**Summary:**

The authors present a new convergence analysis for Decentralized SGD, finding that all eigenvalues of the mixing matrix affect the convergence rate. This analysis leads to the key insight that in homogeneous settings, the effect of topology on the convergence rate is notably smaller than expected from existing analyses, which primarily rely on the spectral gap.

**Strengths:**

* The motivation is strong. The authors address a well-known and long-standing discrepancy in the decentralized learning community: the empirical observation that the spectral gap has little experimental impact on training behavior, despite theoretical analyses suggesting otherwise.
* The core finding that all eigenvalues of the mixing matrix affect the convergence rate is both interesting and novel. It offers a more precise characterization of the algorithm's dynamics.
* The paper is very well-written. The motivation and background are clearly articulated, the new theoretical results are clearly contrasted with prior work (e.g., in Table 1), and the inclusion of a clear proof sketch (Section 5) is helpful for understanding the main technical ideas.

**Weaknesses:**

No major weaknesses identified. Some potential limitations are listed below.

1. The paper's core motivation and main contribution are focused on the homogeneous case. This may limit the contribution to "near-homogeneous" settings. The analysis explains why sparse graphs (like rings) perform well on homogeneous data (as the $\sigma^2$ term is improved), but it also confirms that the spectral gap $p$ remains the critical bottleneck for heterogeneous data ($\zeta > 0$). This remains a limitation, as data heterogeneity is the important in modern decentralized and federated learning.

2. There is a mismatch between the scope of the theory and the experiments. The theory (e.g., Theorem 1) covers both convex and non-convex cases. However, the experiments in Section 6.2 are only run on two convex models (Logistic Regression and Ridge Regression). The paper is missing an empirical validation of its non-convex analysis. Additionally, the experiments are only on MNIST, which is a relatively simple dataset.

**Questions:**

1.  Assumptions 6 and 7 (bounding noise and heterogeneity at the optimum $x^*$) are not standard in the decentralized learning literature. Can the authors provide more detail on their motivation, reasonableness, and specifically where they are critically used in the proof for the convex case?
2.  The paper discusses Neglia et al. (2020) and Vogels et al. (2022). Could the authors provide a more direct and detailed comparison of the final bounds/rates and the underlying assumptions between this work and those two papers?

3.  Regarding the experimental methodology described in Section 6.2 and detailed in Appendix E: The process appears to involve factoring a base mixing matrix $W$ into $W = V \Lambda V^{-1}$, and then reconstructing a new matrix $W' = V \tilde{\Lambda} V^{-1}$ by altering the eigenvalues $\lambda_3, ..., \lambda_n$ while keeping $\lambda_2$ (and thus $p$) constant. What was the main motivation for this synthetic approach? An alternative might be to directly compare the performance on different standard sparse graphs (e.g., Ring vs. Torus), which would naturally have different values for both $p$ and the new metric $\frac{1}{n}\sum_{i=2}^{n}(\lambda_{i}^{2}/1-\lambda_{i}^{2})$.

---

> ### Author Response · Authors · 2025-11-26
>
> Thank you for the reviewer’s positive assessment. We are pleased to hear that no major weaknesses were identified. We addressed the comments in the following.
>
> > The paper's core motivation and main contribution are focused on the homogeneous case. [...]
>
> Thank you for carefully reviewing our paper.
> As the reviewer pointed out, our analysis implies that it is inevitable that the rate depends on the spectral $p$ when $\zeta \gg 0$.
> However, many papers have observed that data heterogeneity does not significantly impact practical performance, e.g., [1], as real datasets are not typically too heterogeneous.
> We therefore believe that understanding the convergence behavior in the near-homogeneous case is also important.
>
> > There is a mismatch between the scope of the theory and the experiments. The theory (e.g., Theorem 1) covers both convex and non-convex cases. However, the experiments in Section 6.2 are only run on two convex models (Logistic Regression and Ridge Regression). [...]
>
> We added the results with CIFAR-10 in Sec. G, and the observation was consistent with the results of MNIST shown in Figure 2.
> Since the results for ridge regression are still in progress, we will add them in the camera-ready version once they are completed.
>
> The performance of the neural networks is typically affcted by many factors, such as generalization ability.
> Our experiments are primary designed to verify the improvement in our new analysis.
> Thus, to keep the experimental setting simple, we focused on the logistic regression and ridge regression in our numerical evaluations.
>
> > Assumptions 6 and 7 (bounding noise and heterogeneity at the optimum $x^\star$) are not standard in the decentralized learning literature.
>
> The standard assumptions used in decentralized literature are the uniform bounded noise, and heterogeneity, given in our assumptions 3 and 4. We use instead Assumptions 6 and 7 (similarly to the prior works: [1]) with bounding noise and heterogeneity at the optimum, as it is **weaker** than commonly used assumptions 3 and 4 in the literature. In fact $\sigma_{\star} \leq \sigma$ and $\zeta_{\star} \leq \zeta$. We believe that using a weaker assumptions only strengthens our results.
>
> > The paper discusses Neglia et al. (2020) and Vogels et al. (2022). Could the authors provide a more direct and detailed comparison of the final bounds/rates and the underlying assumptions between this work and those two papers?
>
> In Sec. A, we incorporated the convergence results of Neglia et al. (2020) and Vogels et al. (2022).
> This addition makes it clearer that our analysis offers a better analysis compared to these prior works.
> Thank you for the insightful comment.
>
> The convergence rate shown in Neglia et al., (2020) requires additional assumptions that the stochastic gradient is bounded, i.e., $\| \nabla F_i (x ; \xi_i) \| \leq G$, to ensure the convergence.
> The convergence rate shown in Vogels et. al., (2020) depends on $n_{W} (\gamma)$ instead of the spectral gap.
> As $n_{W} (\gamma)$ depends on the hyperparameter $\gamma$, it is unclear whether this rate is better than the rate derived in Koloskova et al. (2020).
> For instance, Figure 2 in Vogels et. al., (2020) shows that $n_{W} (\gamma) \ll n$ when $\gamma \approx 0$, which implies that the rate does not achieve linear speedup if we set $\gamma \approx 0$ and is worse than the rate shown in Koloskova et al. (2020).
>
>
> > Regarding the experimental methodology described in Section 6.2 and detailed in Appendix E: [...]
>
>
> The experimental setting is designed to evaluate how the convergence behavior is affected by the new quantity $\frac{1}{n} \sum_{i=2}^n \frac{\lambda_i^2}{1 - \lambda_i^2}$, while keeping the spectral gap $p$ fixed.
> If both $p$ and $\frac{1}{n} \sum_{i=2}^n \frac{\lambda_i^2}{1 - \lambda_i^2}$ change at the same time, it will be hard to interpret from the figures to what extent each of the two quantities affects convergence.
>
>
> ## Reference
>
> [1] Koloskova et al., A unified theory of decentralized SGD with changing topology and local updates, In ICML 2020

---

### Author Response · Authors · 2025-11-28

Dear Reviewers,

We sincerely appreciate your valuable feedback.
We would like to kindly ask whether there are any additional questions or clarifications needed regarding our rebuttal. The deadline of the discussion phase is approaching.
If reviewers have remaining concerns or questions, we would be very happy to address them.

Best,

Authors

---

### Meta-Review · Area_Chair_7TtV · 2025-12-11

**Summary:**

The reviewers indicated the following concerns:
- The improvement only happens for the homogeneous case
- The experiments are not sufficient: the paper only considers small datasets, convex problems and fixed topologies, the paper does not present experiments to justify the performance of Decentralized SGD under different topologies in both IID and non-IID settings
- difference between the spectral gap and full spectral is not intuitive
- a lower bound is missing to justify the tightness of the result in the iid case
- the case of strongly convex case is not covered
- analysis based on eigenvalues requires undirected graphs
- presentation needs improvement as the insight/challenge of the analysis is not clear

**Reviewer Concerns:**

From my understanding, the following concerns are well addressed:

- the case of strongly convex case is not covered (the revision includes the analysis with strongly convex cases)
- difference between the spectral gap and full spectral is not intuitive (the authors' response provides some intuitive understanding on the difference)
- The concerns on the experiments are partially addressed (the revision includes experimental results for nonconvex models and larger datasets)

From my understanding, the following concerns are not quite well addressed:
- The improvement only happens for the homogeneous case. The authors only mentioned that the heterogeneity does not significantly impact practical performance. However, three reviewers (suPG, fzn7, nnfv)  indicate that data heterogeneity is the important in modern decentralized and federated learning.
- a lower bound is missing to justify the tightness of the result in the iid case (the authors simply mentioned that developing lower bounds are beyond the scope of the paper)
- Experimental results under different topologies
- analysis based on eigenvalues requires undirected graphs (the authors mention that it is possible to extend it to directed graphs without giving convincing arguments)

**Reviewer Scores:**

After reading the authors' responses, I think Reviewers suPG, fzn7, nnfv would not change their score since there main concern on the limitation of the improvement in the homogeneous case is not well addressed. Reviewer 2mdZ is likely to change the score from 2 to 4 since his/her concerns on the experimental results and strongly convex analysis are well addressed. Reviewer CSwe may not change the score as   his concern on the extension to directed graph is not well addressed.

---

### Decision · Program_Chairs · 2026-01-26

Reject